# KernelFoundry: Hardware-Aware Evolutionary GPU Kernel Optimization

Nina Wiedemann [* 1]   Quentin Leboutet [* 1]   Michael Paulitsch [1]   Diana Wofk [1]   Benjamin Ummenhofer [1]

## Abstract

GPU kernel optimization challenges LLMs beyond standard coding tasks, as it requires an understanding of hardware architecture, parallel computing optimization strategies, and profiling outputs. However, most existing approaches leveraging LLMs for kernel generation apply standard prompting and feedback loops, considering hardware only through profiling feedback. We introduce KernelFoundry, an evolutionary framework that efficiently explores the space of GPU kernels through (1) MAP-Elites quality-diversity search with kernel-specific behavioral dimensions to sustain exploration; (2) meta-prompt evolution that co-evolves prompts with kernels to uncover task-specific optimization strategies, and (3) a template-based parameter optimization approach to tune kernels to inputs and hardware. We evaluate this framework on *Kernel-Bench*, *robust-kbench* and custom tasks, generating SYCL kernels as a cross-platform GPU programming paradigm, and CUDA kernels for comparison to prior work. Our approach consistently outperforms the baseline methods and achieves an average speedup of 2.3 on KernelBench for SYCL. Moreover, KernelFoundry is implemented as a distributed framework with remote access to diverse hardware, allowing quick benchmarking and featuring a flexible user input layer to support kernel generation for a wide range of real use cases beyond benchmarking.

## 1. Introduction

Writing high-performance GPU kernels requires deep expertise in hardware architectures, memory hierarchies, and parallel programming paradigms: knowledge that remains scarce even among experienced engineers (Li et al., 2025c).

Yet, efficient GPU implementations are often decisive for a method's success and adoption, as illustrated by FlashAttention (Dao et al., 2022), which was instrumental to the practical scalability of Transformers. Following the release of KernelBench (Ouyang et al., 2025b), a growing body of work has explored automating GPU kernel development using LLMs. With few exceptions based on model fine-tuning (Li et al., 2025c; Baronio et al., 2025), most approaches rely on an iterative generate→verify→measure loop: kernels proposed by an LLM are compiled, validated, benchmarked, and the resulting feedback is used to steer subsequent generations. Variants of this paradigm have been demonstrated across CUDA (Chen et al., 2025b; Zhang et al., 2025; Lange et al., 2025b), Triton (Wang et al., 2025; Li et al., 2025a), and Metal (Sereda et al., 2025), often augmented with profiling tools (Zhang et al., 2025; Chen et al., 2025b; Lange et al., 2025b; Sereda et al., 2025), multi-agent architectures (Wei et al., 2025; Zhang et al., 2025; Lei et al., 2025), or evolutionary search techniques (Liao et al., 2025; Lange et al., 2025a; Yan et al., 2026). Despite known shortcomings in KernelBench that necessitate careful interpretation of reported speedups (Lange et al., 2025b), these studies collectively demonstrate that LLMs can produce correct and performant kernels when guided by execution feedback. However, existing approaches rely on general code-generation strategies that fail to address the unique challenges of kernel optimization. Unlike standard programming tasks, kernel generation demands (1) knowledge of memory access patterns and an understanding of how parallelism maps to specific architectures, and (2) a systematic exploration of optimization strategies. Furthermore, iterative prompting and kernel validation loops suffer from *mode collapse* – iterative refinement behaves as greedy local search, with LLMs repeatedly proposing variants close to recent successes, leading to premature convergence – and *context degradation* – as optimization histories grow, failed attempts dominate the prompt context.

To overcome these limitations, we propose *KernelFoundry*, an evolutionary framework that maintains a diverse archive of kernel implementations and systematically explores optimization techniques. To counteract mode collapse, KernelFoundry adapts a MAP-Elites evolutionary strategy (Mouret & Clune, 2015). Rather than defining generic behavioral descriptors such as code length (Lehman

[1]Intel Corporation * Equal contribution. Correspondence to: Nina Wiedemann <nina.wiedemann@intel.com>.

*Proceedings of the 43rd International Conference on Machine Learning*, Seoul, South Korea. PMLR 306, 2026. Copyright 2026 by the author(s).

et al., 2023), kernel implementations are indexed along well-defined, *domain-specific dimensions addressing hardware system architecture*, such as memory access patterns or parallelism strategies. To mitigate context degradation, we leverage *meta-prompt evolution*, allowing prompts to co-evolve with kernels and uncover task-specific optimization strategies. Finally, we propose to repurpose *templated kernels* to allow the LLM to tune hardware-dependent values like tile and block sizes.

We evaluate KernelFoundry on KernelBench, robust-kbench, and custom benchmark suites that span single-kernel operators, fusion patterns, and full model architectures. Unlike prior work that targets NVIDIA's proprietary ecosystem almost exclusively, we additionally use SYCL, an open-standard C++ abstraction that is vendor-agnostic and more expressive than other cross-platform GPU programming frameworks such as Triton. Furthermore, our framework is not limited to standardized tasks or benchmarks. We introduce a custom input format that allows users to optimize kernels based on arbitrary instructions (not restricted to PyTorch code) and to employ complex test frameworks, provided they are compatible with pytest. As a case study, we showcase the optimization of a specific operation within a LLama3 model, illustrating the practical utility of our pipeline for real-world, model-level tasks.

**Conflict of Interest Disclosure.** The authors declare no financial conflicts of interest related to this work.

## 2. Related Work

**Traditional Kernel Optimization.** High-performance computing has long relied on separating algorithm specification from optimization schedules, as exemplified in Halide (Ragan-Kelley et al., 2013) or TVM (Chen et al., 2018a), whereas lower-level models like CUDA and SYCL require both to be handled together in code. Triton (Tillet et al., 2019) offers a middle ground with a Python-based DSL that simplifies custom GPU kernel development but still exposes scheduling control. Vendor libraries (cuDNN (Chetlur et al., 2014), CUTLASS (NVIDIA Corporation, 2017), oneDNN (Intel Corporation, 2016)) deliver peak performance but lack flexibility for novel operators. Classical autotuners like ATLAS (Whaley et al., 2001) and FFTW (Frigo & Johnson, 1998) demonstrate that systematic search can match hand-tuned code. To avoid exhaustively evaluating every schedule, systems like AutoTVM (Chen et al., 2018b), AnsorAnsor (Zheng et al., 2020), and Ithemal (Mendis et al., 2019) use learned cost models to predict performance, but these require extensive training data and struggle to generalize across hardware. Despite these tools, algorithmic kernel development remained a highly manual and expertise-driven process.

**LLM-Based Kernel Generation.** Large language models have sparked intense interest in automated kernel synthesis, yet generating *correct* GPU code remains challenging, and generating *fast* code is harder still. Early studies (Valero-Lara et al., 2023; Godoy et al., 2023) evaluated Llama-2 and Codex on generating HPC kernels. The introduction of KernelBench (Ouyang et al., 2025b) established the de facto benchmark, comprising 250 tasks spanning single operators, fusion patterns, and complete architectures. Results showed that even frontier models achieve correctness on merely ∼20% of problems with substantial gaps to cuBLAS/cuDNN baselines. Recent work has also highlighted validation challenges: spurious speedups of up to 50x can arise from incorrect kernels that pass test cases (Lange et al., 2025b; Zhu et al., 2025), motivating more rigorous evaluation protocols. Despite these challenges, KernelBench has inspired a plethora of follow-up approaches (see Table 5 for a comprehensive overview). The dominant paradigm is *prompting with validation loop*: correctness and profiling outputs guide successive refinements of CUDA (Chen et al., 2025b; Zhang et al., 2025; Lange et al., 2025b), Triton (Liao et al., 2025; Li et al., 2025a), and Metal (Sereda et al., 2025) kernels. PEAK (Tariq et al., 2025) expresses optimization strategies as natural language transformations, achieving up to 95% of cuBLAS performance on MatMul across CUDA, HIP, and HLSL. Multi-agent architectures decompose the task into specialized roles: Astra (Wei et al., 2025) separates planning, coding, testing, and profiling agents; STARK (Dong et al., 2025) uses plan-code-debug agents; QiMeng (Zhu et al., 2026) employs macro/micro coding agents; and PRAGMA (Lei et al., 2025) coordinates coder, verifier, and conductor roles. SwizzlePerf (Tschand et al., 2025) stands out by injecting detailed microarchitecture knowledge (bank conflicts, warp operations, coalescing) into prompts, demonstrating the value of hardware awareness. Despite their sophistication, these approaches share a common limitation: they explore the kernel design space implicitly through generic code generation, without mechanisms to preserve diversity across optimization trajectories.

An orthogonal direction is *finetuning* LLMs for kernel generation. CUDA-L1 (Li et al., 2025c) combines supervised fine-tuning (SFT) with "contrastive RL"; CUDA-L2 (Su et al., 2025) adds NCU profiler metrics and multi-stage RL, exceeding cuBLASLt by 11% on HGEMM, TritonRL (Woo et al., 2025) presents good results with SFT and GRPO for Triton, and Kevin (Baronio et al., 2025) uses multi-turn RL to refine kernels across dialogue turns. These methods achieve promising results but often cannot match the performance of the latest closed-source LLMs.

Existing LLM kernel work targets CUDA almost exclusively. MultiKernelBench (Wen et al., 2025) addresses this with benchmarks spanning NVIDIA GPUs, Huawei NPUs, and

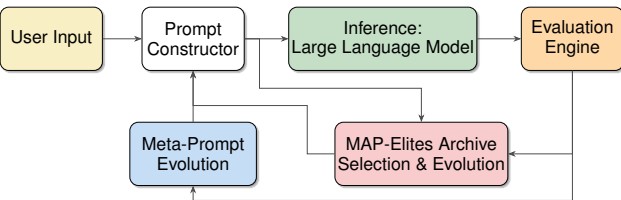

*Figure 1.* The KernelFoundry pipeline: evolutionary kernel optimization with multi-level feedback and meta-prompt co-evolution.

Google TPUs, and NPUEval (Kalade & Schelle, 2025) provides AMD NPU benchmarks. Translation systems like BabelTower (Wen et al., 2022) and CodeRosetta (TehraniJamsaz et al., 2024) convert between CUDA and HIP/OpenCL but preserve rather than optimize performance. Our work contributes to this direction by targeting SYCL (Keryell et al., 2015) and hence enabling portability across Intel, NVIDIA, and AMD accelerators.

**LLM-Guided Evolutionary Search.** Evolutionary algorithms have a rich history in program synthesis (Koza, 1992). Quality-diversity (QD) methods (Pugh et al., 2016) shift from single-objective optimization to maintaining *diverse collections* of high-performing solutions: a property particularly relevant for kernel optimization, where multiple valid implementation strategies exist. MAP-Elites (Mouret & Clune, 2015) partitions the solution space by behavioral descriptors, CMA-ME (Fontaine et al., 2020) adds continuous optimization, and PGA-MAP-Elites (Fontaine & Nikolaidis, 2021) integrates policy gradients. Recent work combines LLMs with evolutionary search: FunSearch (Romera-Paredes et al., 2024) discovers mathematical programs beyond human baselines, AlphaEvolve (Novikov et al., 2025) adds multi-objective optimization, and CodeEvolve (Assumpção et al., 2025) evolves prompts rather than code directly. PACEvolve (Yan et al., 2026) characterizes three failure modes in LLM-assisted evolution: *context pollution* from accumulated failures, *mode collapse* from poor exploration-exploitation balance, and *weak collaboration* in multi-island settings. While PACEvolve addresses these through hierarchical context management for general search, kernel optimization presents a *structured* space where domain-specific dimensions can be directly exploited. Our approach leverages this structure: kernel-specific behavioral coordinates (memory patterns, parallelism strategies) inherently partition the solution space, preventing mode collapse by construction, while meta-prompt evolution (Agrawal et al., 2025) counteracts context degradation by enabling prompts to specialize for different optimization regions.

# 3. Method

## 3.1. System Architecture

We present KernelFoundry, an evolutionary framework for LLM-driven GPU kernel optimization. Figure 1 illustrates the architecture, which comprises five tightly-coupled components. The process begins with a **task specification layer** that accepts kernel generation problems in a flexible format, supporting PyTorch reference implementations, high-level natural language descriptions, or existing kernels requiring optimization. The system natively supports KernelBench (Ouyang et al., 2025b) tasks as well as custom workloads with user-defined test frameworks. A **prompt construction engine** then assembles context-aware prompts by combining the task specification with hardware-specific optimization guidance, sampled parent programs and inspirations from the archive, gradient-derived mutation hints, and dynamically evolved prompt components (§3.5). The resulting prompt is served to an **LLM inference backend**, which provides a unified interface to both API-based models (OpenAI, Anthropic) and locally-hosted models via vLLM, and generates multiple candidate kernels. Candidates are processed by a **compilation & evaluation pipeline** that compiles to the target backend (SYCL via Intel DPC++, CUDA via nvcc, or Triton), validates numerical correctness against reference implementations, measures execution time, and optionally collects profiling data via Intel *unitrace* or NVIDIA *Nsight*. Finally, an **evolutionary archive** implements MAP-Elites (Mouret & Clune, 2015) by organizing kernels according to behavioral coordinates in an optimization feature space (§3.2). It retains the best-performing kernel (elite) for each occupied cell, thereby preserving diversity across qualitatively different optimization strategies. In a similar spirit, we evolve sections of the prompt with **meta-prompt evolution**, which maintains a separate archive with successful guidance examples.

The evolutionary loop proceeds as follows. At each iteration, parent programs are sampled from the archive using selection strategies informed by fitness, exploration potential, and gradient estimates. The prompt constructor assembles a generation prompt, incorporating parent code, feedback from prior evaluations, and optimization guidance. The LLM generates candidate kernels, which are then compiled, correctness-tested, and performance-measured. Candidates that run correctly are assigned behavioral coordinates via static code analysis; those improving upon existing elites are inserted into the archive. Feedback from all outcomes (including failures) informs subsequent iterations through the gradient estimator and meta-prompter.

## 3.2. Quality-Diversity Search with MAP-Elites

Standard evolutionary approaches optimize a single objective (e.g., runtime), risking convergence to local op-

tima and failing to explore alternative solution strategies. Quality-Diversity (QD) algorithms (Pugh et al., 2016) address this limitation by maintaining diverse collections of high-performing solutions in independent behavioral cells, enabling discovery of stepping-stone solutions that facilitate escape from local optima.

**MAP-Elites Algorithm.** MAP-Elites (Mouret & Clune, 2015) partitions the solution space into a discrete grid based on *behavioral descriptors*, which are low-dimensional features characterizing solution behavior independent of fitness. Each cell maintains the highest-fitness solution discovered for that behavioral region. The algorithm iterates in four phases (**selection**, **variation**, **evaluation**, and **insertion**). During **selection**, the system samples parent kernel(s) from the archive using a configurable strategy (uniform, fitness-proportionate, curiosity-driven, or island-based). During **variation**, it produces offspring via LLM-based code modification guided by the parent kernel, optimization hints, and meta-evolved prompt content. During **evaluation**, the offspring is compiled, validated for correctness, benchmarked for performance, and assigned behavioral coordinates via static code analysis. During **insertion**, the offspring replaces the incumbent elite if it improves upon the existing elite (or if the target cell is empty); otherwise, it is discarded. This mechanism maintains diversity by construction: the archive cannot collapse because each cell evolves independently.

**Kernel-Specific Behavioral Descriptors.** Adapting the MAP-Elites algorithm to kernel optimization requires defining behavioral descriptors that capture meaningful optimization characteristics. With KernelFoundry, we propose a three-dimensional optimization feature space with discrete levels reflecting increasingly sophisticated optimization patterns. These coordinates are computed deterministically from generated code via static pattern matching on SYCL and CUDA constructs, ensuring reproducibility and reducing execution-time variability.

1. **Memory Access Pattern** ($d_{\mathrm{mem}} \in \{0, 1, 2, 3\}$):
   **0:** Scalar, strided, or uncoalesced access
   **1:** Coalesced/vectorized (vec4, aligned loads)
   **2:** Shared/local memory with explicit tiling
   **3:** Multi-lev. hierarchy (SLM + reg. blocking + prefetch)

2. **Algorithmic Structure** ($d_{\mathrm{algo}} \in \{0, 1, 2, 3\}$):
   **0:** Direct PyTorch translation
   **1:** Fused operations (single-pass over data)
   **2:** Reformulated algorithm (online norm., flash pattern)
   **3:** Novel/asymptotically improved algorithm

3. **Parallelism Coordination** ($d_{\mathrm{sync}} \in \{0, 1, 2, 3\}$):
   **0:** No synchronization (embarrassingly parallel)
   **1:** Work-group barriers (`group::barrier`)
   **2:** Sub-group primitives (shuffles, reductions, broadc.)

**3:** Global coordination (atomics, multi-pass with sync)

This three-dimensional grid yields $4^3 = 64$ behavioral cells. The classifier employs weighted pattern matching with category-specific patterns to avoid double-counting related constructs (e.g., a kernel using *group_barrier* for SLM synchronization receives credit in $d_{\mathrm{mem}}$ for SLM usage, not additionally in $d_{\mathrm{sync}}$ for the same barrier).

**Fitness Function.** The fitness function prioritizes correctness while rewarding performance:

$$f(k) = \begin{cases} 0 & \text{if compilation fails,} \\ 0.1 & \text{if compiles but incorrect,} \\ 0.5 + 0.5 \cdot s_{\mathrm{norm}} & \text{if correct,} \end{cases}$$

where $s_{\mathrm{norm}} = \min(1, \mathrm{speedup}/\mathrm{target})$ is the normalized speedup relative to a configurable target (default: $2\times$ over PyTorch baseline). This design ensures that correctness is a prerequisite for high fitness while providing a continuous gradient for performance optimization.

**Selection Strategies.** We implement four parent selection strategies, with configurable mixing ratios. Uniform selection samples randomly from occupied cells to maximize behavioral diversity. Fitness-proportionate selection weights sampling by elite fitness to exploit high-performing regions. Curiosity-driven selection prioritizes cells with high estimated improvement potential as inferred from the gradient signal. Island-based selection maintains $K$ independent subpopulations with periodic migration every $M$ generations, balancing isolated exploration with cross-pollination.

### 3.3. Gradient-Informed Evolution

While the archive structure prevents global mode collapse, individual cells may stagnate if the LLM repeatedly produces similar variants. Inspired by CMA-ME (Fontaine et al., 2020) and PGA-MAP-Elites (Fontaine & Nikolaidis, 2021), we augment MAP-Elites with gradient-like signals derived from evolutionary transition history. These gradients provide a *soft steering mechanism* that complements the archive's guarantees of structural diversity.

**Transition Tracking and gradient estimation.** We maintain a circular buffer of recent parent→child transitions. Each transition record stores the parent and child behavioral coordinates $(\mathbf{b}_p, \mathbf{b}_c)$, the fitness delta $\Delta f = f_c - f_p$, the transition outcome (improvement when the child becomes an elite or discovers a new cell, neutral when it is competitive but does not update the archive, or regression when fitness decreases), as well as a timestamp and iteration number for temporal weighting.

From the accumulated transitions, we compute three gradient components for each occupied cell $\mathbf{b}$. The *Fitness gradient* $\nabla_{\mathbf{b}} F$ estimates which directions in behavioral space

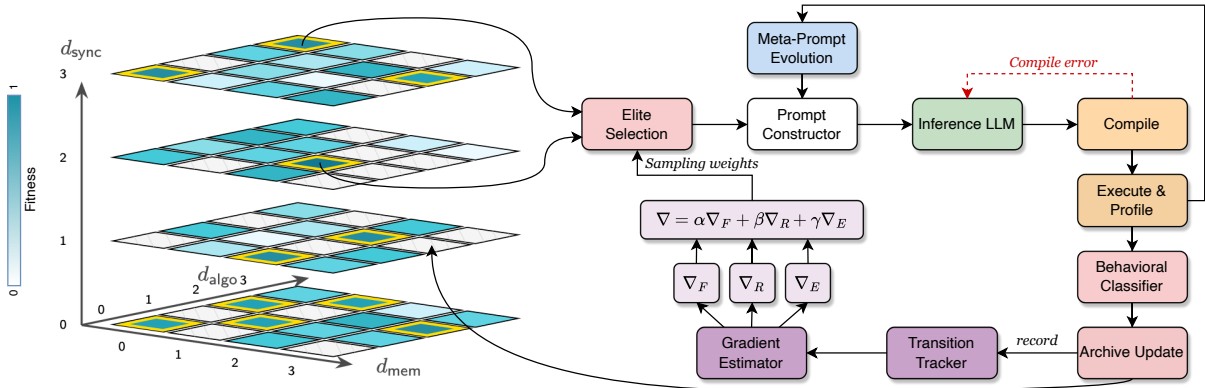

*Figure 2.* Gradient-informed MAP-Elites for kernel optimization. The archive partitions kernels by behavioral coordinates $(d_{\text{mem}}, d_{\text{algo}}, d_{\text{sync}})$. Elites are shown in yellow. A Transition Tracker records parent→child transitions with behavioral coordinates and fitness deltas. The Gradient Estimator combines fitness gradients ($\nabla F$), improvement-rate gradients ($\nabla R$), and exploration gradients ($\nabla E$) to produce sampling weights and natural-language mutation hints that guide subsequent generations.

improve fitness. For each dimension $d$, we aggregate transitions weighted by fitness delta and direction:

$$\nabla_d F \approx \frac{1}{|\mathcal{T}|} \sum_{t \in \mathcal{T}} \Delta f_t \cdot \text{sign}(b_c^{(d)} - b_p^{(d)}) \cdot w(t) \quad (1)$$

where $\mathcal{T}$ is the set of transitions originating from cell $\mathbf{b}$, and $w(t)$ applies exponential time decay to prioritize recent experience. *Improvement-rate gradient* $\nabla_\mathbf{b} R$ estimates which directions yield higher improvement probability, independent of improvement magnitude:

$$\nabla_d R \approx P(\text{imp.} \mid \Delta b_d > 0) - P(\text{imp.} \mid \Delta b_d < 0) \quad (2)$$

*Exploration gradient* $\nabla_\mathbf{b} E$ points to empty or low-quality cells, weighted by inverse distance and improvement potential:

$$\nabla_\mathbf{b} E \propto \sum_{c \in \mathcal{E}} \frac{(f_{\max} - f_c)}{\|\mathbf{c} - \mathbf{b}\|_1} \cdot \frac{\mathbf{c} - \mathbf{b}}{\|\mathbf{c} - \mathbf{b}\|_1} \quad (3)$$

where $\mathcal{E}$ is the set of empty cells and low-quality occupied cells. The combined gradient is a weighted average, $\nabla_\mathbf{b} = \alpha \nabla F + \beta \nabla R + \gamma \nabla E$, with $(\alpha, \beta, \gamma) = (0.4, 0.4, 0.2)$.

**Gradient-to-Prompt Translation.** Gradients inform the optimization process at two levels. For *parent selection*, cells with strong positive gradient magnitudes receive higher sampling probability, directing computational effort toward productive regions. For *prompt construction*, gradient directions are translated into natural-language mutation hints. For example, a positive gradient in $d_{\text{mem}}$ yields hints such as "*consider adding shared memory tiling*" or "*implement register blocking for data reuse*." These hints are injected into the generation prompt, providing actionable guidance grounded in empirical transition success (see example in Appendix E.3).

### 3.4. Templated Kernels for Parameter Tuning

Beyond algorithmic transformations, kernel performance depends critically on hardware-specific parameters such as work-group dimensions, tile sizes, and unroll factors. Rather than requiring the LLM to guess optimal values, we enable explicit parameter exploration through templated kernel generation. The prompt instructs the LLM to optionally produce a *templated kernel* with configurable parameters alongside a dispatch function enumerating valid parameter combinations. Our evaluation pipeline detects templated kernels, extracts parameter configurations, and evaluates each instantiation independently. The best-performing configuration determines the kernel's fitness, while all results are logged to enable the LLM to refine parameter choices in subsequent iterations. This approach separates algorithmic optimization from parameter tuning, allowing the evolutionary search to explore both dimensions efficiently.

### 3.5. Meta-Prompting Against Context Degradation

As evolutionary search progresses, accumulated experiment histories can degrade generation quality as failed attempts dominate the context: a phenomenon termed *context pollution* (Yan et al., 2026). Inspired by CodeEvolve (Assumpção et al., 2025) and GEPA (Agrawal et al., 2025), we introduce **meta-prompt evolution**: prompts are mutable and co-evolve with kernels, enabling successful optimization strategies to propagate while unsuccessful guidance is pruned.

**Evolvable Prompt Regions.** The kernel generation prompt contains four evolvable sections, delimited by special markers. Specifically, we evolve (1) **optimization philosophy**, which encodes high-level principles that shape priorities (e.g., "*prioritize memory bandwidth utilization before compute optimization*"), (2) **optimization strategies**,

which enumerate concrete techniques organized by category (memory, compute, parallelism) together with canonical code patterns and short explanations, (3) **common pitfalls**, which list anti-patterns and frequent mistakes to avoid (e.g., "*avoid bank conflicts by adding padding to shared memory arrays*"), and (4) **analysis guidance**, which provides a pre-coding reasoning scaffold that prompts the LLM to identify likely bottlenecks before generating code.

**Meta-Prompter LLM.** This dedicated LLM (distinct from the kernel generator) analyzes generation outcomes and proposes prompt modifications. Given the current evolvable prompt sections together with the generated kernel code and evaluation metrics (correctness, speedup, and error messages), the meta-prompter first diagnoses which guidance was missing, misleading, or insufficiently specific for the observed outcome. It then prescribes targeted updates as SEARCH/REPLACE diffs restricted to the evolvable regions. This two-LLM architecture separates analysis from generation, allowing the meta-prompter to specialize in prompt refinement without conflating it with kernel synthesis.

**Prompt Archive and Co-Evolution.** Evolved prompts are maintained in their own archive, with fitness defined by the best kernel performance achieved using each prompt variant. Kernels and prompts co-evolve in an interleaved schedule: every $N$ kernel generations (default $N = 10$), the meta-prompter proposes prompt updates based on recent outcomes. This co-evolutionary dynamic enables the system to discover task-specific optimization strategies that would be difficult to engineer manually.

### 3.6. Distributed Execution Framework for Scalability

Scalability is essential in adoption of kernel generation in practice. To support efficient parallel evaluation across heterogeneous hardware, KernelFoundry employs a distributed architecture with four worker types (see Appendix C Figure 4 for a graphical overview): (1) LLM server, (2) compilation worker, (3) execution workers and (4) database server. While parallelization for (1) is common, separation and distribution of (2) and (3) makes KernelFoundry truly scale. The evaluation loop scales with flexible hardware allocation as only execution workers require GPUs, and users can target different backends (Intel, NVIDIA) by routing jobs to appropriate workers. Additionally, multiple workers with identical hardware can be used to speed up the pipeline to compile and evaluate kernels in parallel.

## 4. Experimental setup

The field of kernel generation faces challenges in making fair comparisons, because results depend heavily on the hardware used, the programming approach, and the abilities of the LLM. To address this, we carefully control these factors and conduct experiments across several benchmarks, using different hardware types and programming languages.

**Programming paradigm and hardware.** KernelFoundry supports CUDA, SYCL, and Triton. Our primary focus is on SYCL for Intel hardware. SYCL offers several advantages: (1) it is an open industry standard (Khronos Group) rather than a proprietary language; (2) it provides C++ abstractions that are more amenable to LLM generation than low-level intrinsics; (3) it enables true cross-platform portability with a single codebase. The experiments are conducted on an integrated Intel Arc 140V GPU and a discrete Intel Battlemage B580 GPU, as well as an NVIDIA RTX A6000 GPU with Ampere architecture (for CUDA comparisons). We implement a rigorous kernel performance benchmarking with several improvements over prior work (see Appendix B).

**Benchmarks.** We evaluate on *KernelBench* (Ouyang et al., 2025b) as the de-facto standard for LLM kernel generation, comprising 250 tasks across three levels (L1: single operators, L2: fusion patterns, L3: full architectures). Following Lange et al. (2025b) and METR (2025), we filter out flawed tasks prone to reward hacking or with inefficient baselines, resulting in a filtered set of 111 tasks (80 L1, 31 L2). For most experiments, we use a representative subset of 40 tasks (20 L1, 20 L2), that is selected based on related work (see Appendix D). Secondly, we utilize 12 tasks from *Robust-kbench* (Lange et al., 2025b). They provide 19 tasks including forward-backward operations; however, they published their results (i.e., the best generated kernel, required for comparison) only for 12 of those.

**Baselines.** Following KernelBench, we compare to Pytorch eager execution and torch.compile. Unfortunately, most related works only publish aggregated numbers such as the average speedup per level, usually across all KernelBench problems, and not the generated kernels themselves. Since some KernelBench tasks are flawed and allow reward hacking, and the performance is highly hardware dependent, comparability to these results is inhibited. We thus compare to prior work that publish their best-performing kernels: robust_kbench (Lange et al., 2025b), the AI CUDA Engineer (Lange et al., 2025a) and Kernelsseum (Ouyang et al., 2025a;b). Due to model deprecation and access restrictions, we are limited to using only the available subset of the LLMs used in the baselines, which presents additional challenges for our method. For comparison with robust-kbench, we utilize GPT-{o3,o4-mini,4.1}, excluding Sonnet 3.7 used in their method. For comparing to the AI CUDA engineer we could only use o3-mini (excluding DeepSeek-{v3,R1}, GPT-{o1-preview,o1-high}, and Sonnet 3.5). Further hyperparameter settings are given in Appendix B.

**Metrics.** Following prior work, we report the correctness rate (fraction of tasks with a kernel that compiles and yields

numerically correct outputs) and the fast$_p$ metric, defined as the proportion of tasks with a speedup greater than p: $\frac{1}{N} \sum_{i=1}^{N} \mathbf{1}[\text{speedup}_i > p]$ where $N$ is the number of tasks and the speedup is the baseline time (pytorch eager if not denoted otherwise) divided by the kernel execution time. For correctness, we apply stricter criteria than prior work since we noticed that the precision used by KernelBench is very low, in particular the absolute precision of $10^{-2}$, allowing erroneous kernels to pass in cases of small output values which commonly appear with AI operations. We thus rely on the relative precision $\nu = \frac{|y-\hat{y}|}{|y|+\epsilon}$, where $y$ is the expected output, $\hat{y}$ is the kernel output, and $\epsilon$ is a small value preventing division by zero. However, due to hardware imprecision, errors should be allowed in a small fraction of cases. Thus, we define a kernel as correct if $\nu < 0.01$ in 99% of the output values. As a second measure for kernel correctness, we propose the cosine similarity of the flattened output tensors as a measure of their angular divergence.

# 5. Results

We first evaluate KernelFoundry's ability to generate CUDA kernels in comparison to prior work, and then demonstrate its versatility by generating SYCL kernels. Furthermore, we showcase its adaptability to custom tasks and analyze hardware-specific aspects.

## 5.1. Baseline comparison

Table 1 compares our method to the kernels published by "AI CUDA Engineer" and the robust-kbench method on the respective task set for which the baseline kernels were available. We reproduce the speedup of these kernels on our NVIDIA hardware for fair comparison; original results are listed for reference but are not directly comparable due to hardware-specific PyTorch execution times. We further include the results originally published within KernelBench ("Kernelsseum"). Table 1 shows that the kernels generated with our approach consistently outperform the baselines, despite using only a subset of the LLMs. Specifically, our approach leads to an average speedup of 1.24 on the Kernel-Bench subset L1 and 2.1 on L2, compared to 1.01 and 1.61 for the kernels published by Lange et al. (2025a). On the 12 robust-kbench tasks which include backward operations, our approach also results in a 36% higher speedup compared to the best kernels published by Lange et al. (2025b). While for many tasks the general evolutionary selection procedure already finds the suitable parameters (e.g. tile size), our parameter optimization, applied only for 2 iterations (best@8), can push the performance in some cases, leading to a notable improvement. For results on a per-task level, see Appendix F. The cost per kernel is around $5.7 which is comparable to related work (see Appendix I).

## 5.2. Evaluation on SYCL kernel generation

Table 2 presents our results for SYCL kernel generation. For comparison, we include robust-kbench CUDA performance on the filtered KernelBench tasks, as reported in (Lange et al., 2025b). Although SYCL is less commonly used and less familiar to LLMs than CUDA, our method finds correct kernels in 97% of cases and achieves an average speedup of 2.32. We further compare to OpenEvolve, the open-source implementation of AlphaEvolve (Novikov et al., 2025), which uses an evolutionary algorithm but lacks kernel-specific optimization strategies, meta-prompting, and parameter optimization. While OpenEvolve achieves a comparable speedup to our method after 40 iterations, it requires substantially more iterations to reach this level, as indicated by the notable performance gap after 10 iterations (see Table 2). Ablation studies with 20 iterations show that the performance is reduced without prompt evolution or without gradient guidance. Figure 3 illustrates how KernelFoundry progressively improves kernel performance. Figure 6 shows two cases where meta-prompting becomes increasingly valuable with prolonged training, yielding consistent improvements over 100 iterations. For reproducibility, we provide results obtained with GPT-OSS in Appendix G.

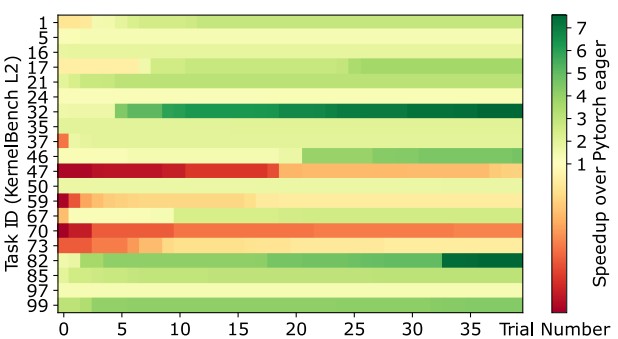

*Figure 3.* Improvement over iterations (cumulative best)

## 5.3. Validating hardware awareness

To analyze whether our method produces hardware-aware kernels, we propose a crossover experiment: We run KernelFoundry on two distinctly different GPUs, here Intel Arc B580 and an integrated Intel Arc 140V GPU (LNL), and then benchmark the performance of the best generated kernels on the respective other GPU. If KernelFoundry is successful in generating hardware-aware kernels, we expect scenarios where the top-performing kernel of the LNL-run outperforms the best kernel optimized on B580 when both are benchmarked on LNL, and vice versa. To quantify this, we introduce the hardware-speedup metric $hws$, defined as the speedup of a kernel $k^A$ optimized on GPU $A$ over a kernel $k_B$ optimized on GPU $B$: $hws(k^A) = \frac{t_A(k^B)}{t_A(k^A)}$, where $t_A$ is the runtime of a kernel on GPU $A$. $hws_p$ is the percentage of kernels for which $hws > p$. We again use

| Task set n = count | Method | LLMs | Correct rate | fast$_1$ | fast$_2$ | Avg. speedup | Geom. speedup |
|---|---|---|---|---|---|---|---|
| KernelBench repr. set L1 n = 20 | Kernelsseum* | GPT-{o1,4o}, DeepSeek-Coder, Sonnet-3.5, LLama-3.1-405B | 0.65 | 15 % | 0 % | 0.765 | 0.600 |
| | AI CUDA Engineer original[†] | DeepSeek-{v3,R1}, GPT-{o1-preview,o1-high,o3-mini}, Sonnet-3.5 | 1.0 | 70 % | 20 % | 1.422 | 1.222 |
| | AI CUDA Engineer re-eval | DeepSeek-{v3,R1}, GPT-{o1-preview,o1-high,o3-mini}, Sonnet-3.5 | 1.0 | 40 % | 50 % | 1.005 | 0.909 |
| | Ours | o3-mini | 1.0 | **55 %** | 10 % | 1.204 | 1.092 |
| | Ours + parameter optim. | o3-mini | 1.0 | **55 %** | 15 % | **1.241** | **1.124** |
| KernelBench repr. set L2 n = 20 | Kernelsseum* | GPT-{o1,4o}, DeepSeek-Coder, Sonnet-3.5 | 0.65 | 10 % | 0 % | 0.874 | 0.854 |
| | AI CUDA Engineer original[†] | DeepSeek-{v3,R1}, GPT-{o1-preview,o1-high,o3-mini}, Sonnet-3.5 | 1.0 | 100% | 10 % | 1.589 | 1.524 |
| | AI CUDA Engineer re-eval | DeepSeek-{v3,R1}, GPT-{o1-preview,o1-high,o3-mini}, Sonnet-3.5 | 1.0 | 85 % | 25 % | 1.606 | 1.515 |
| | Ours | o3-mini | 1.0 | **90 %** | 40 % | 2.051 | 1.680 |
| | Ours + parameter optim. | o3-mini | 1.0 | **90 %** | 45 % | **2.104** | **1.730** |
| Robust-kbench n = 12 | Robust-kbench original[†] | GPT-{o3,o4-mini,4.1}, Sonnet-3.7 | 1.0 | 92 % | 50 % | 15.622 | 2.591 |
| | Robust-kbench re-eval | GPT-{o3,o4-mini,4.1}, Sonnet-3.7 | 1.0 | 92 % | 58 % | 8.865 | 3.499 |
| | Ours | GPT-{o3,o4-mini,4.1} | 1.0 | **100 %** | **67 %** | **12.107** | **5.295** |
| | Ours + parameter optim. | GPT-{o3,o4-mini,4.1} | 1.0 | **100 %** | **67 %** | 12.081 | 5.225 |

*Table 1.* Comparison to baselines on CUDA kernel optimization. Best results per task set in bold. *Kernelsseum* evaluated on Nvidia L40S. [†]*AI CUDA Engineer original* and *Robust-kbench original* both evaluated on Nvidia H100. All others evaluated on Nvidia A6000.

| Task set n = count | Method | Language | LLMs | Correct rate | fast$_1$ | fast$_2$ | Avg. speedup | Geom. speedup |
|---|---|---|---|---|---|---|---|---|
| KernelBench filtered n = 111 | Ours | SYCL | GPT-{4.1, 5-mini}, Sonnet-4.5 | 0.97 | 71 % | 42 % | **2.32** | 1.38 |
| | Robust-kbench | CUDA | GPT-{o3, o4-mini, 4.1}, Sonnet-3.7 | - | - | - | 1.49 | - |
| KernelBench repr. set L2 n = 20 | OpenEvolve (40 iters) | SYCL | GPT-{4.1, 5-mini}, Sonnet-4.5 | 1.0 | 70 % | 40 % | 2.535 | 1.775 |
| | Ours (40 iters + param. optim.) | SYCL | GPT-{4.1, 5-mini}, Sonnet-4.5 | 1.0 | **80 %** | 45 % | **2.732** | **2.119** |
| | OpenEvolve (10 iters) | SYCL | GPT-{4.1, 5-mini}, Sonnet-4.5 | 1.0 | 70 % | 20 % | 1.483 | 1.202 |
| | Ours (10 iters) | SYCL | GPT-{4.1, 5-mini}, Sonnet-4.5 | 1.0 | **80 %** | 35 % | 2.059 | 1.661 |
| | Ours (20 iters) | SYCL | GPT-{4.1, 5-mini}, Sonnet-4.5 | 1.0 | 70 % | 50 % | 2.539 | 1.833 |
| | Ours (20 iters) w/o prompt evolution | SYCL | GPT-{4.1, 5-mini}, Sonnet-4.5 | 1.0 | 70 % | **55 %** | 2.505 | 1.600 |
| | Ours (20 iters) w/o gradient guidance | SYCL | GPT-{4.1, 5-mini}, Sonnet-4.5 | 1.0 | 75 % | 35 % | 2.178 | 1.695 |

*Table 2.* Evaluating KernelFoundry on SYCL kernel generation.

the representative subset of KernelBench L2 (20 tasks) and report average $hws$ for all kernels and $hws_p$ for $p$ equal to 1.0 and 1.5, i.e. the relative amount of kernels being quicker and 1.5 times quicker, respectively.

| Kernels | $hws_1$ | $hws_{1.5}$ | avg. hws | geom. hws |
|---|---|---|---|---|
| LNL-optimized $k^{LNL}$ | 70 % | 20 % | 1.537 | 1.297 |
| BMG-optimized $k^{B580}$ | 70 % | 15 % | 1.109 | 1.038 |

*Table 3.* Evaluating hardware awareness. Kernels optimized for the target GPU outperform kernels optimized on another GPU.

As shown in Table 3, 70% of kernels are faster than their counterpart optimized on another GPU, with average speedup of 1.537 for LNL-optimized kernels and 1.109 for BMG-optimized kernels. See Appendix F.5 for per-task results.

### 5.4. Beyond Pytorch Eager - comparison to oneDNN

KernelBench and follow-ups usually just use Pytorch Eager and torch.compile as baselines, which is a good common ground but adds additional factors, like CPU overhead for routing to specific kernels of a backend library or the effectiveness of the compiled graph. To eliminate these factors and demonstrate the flexibility of our framework beyond the KernelBench format, we directly compare to the high performance kernel library oneDNN. We benchmark our framework on the operations shown in Table 4 implementing them with the C++ API of oneDNN and use operator fusion where possible. For the *concat(x, layernorm(x))* operation we use an initial implementation as starting point using our framework for kernel optimization. For the softmax operation we provide additional high level user guidance to implement an optimization idea that reduces the load on special function units inspired by Flash Attention 4. Table 4 shows that although the kernels in oneDNN are highly optimized, oftentimes implemented directly in assembly language, the generated SYCL kernels are competitive and provide a speedup in three cases.

| Operation | Initial impl. | User instructions | Speedup |
|---|---|---|---|
| concat(x, layer_norm(x)) | X | | 1.79 |
| Matmul with relu post-op | | | 0.35 |
| MaxPool + Linear | | | 0.72 |
| Sum Reduction | | | 1.10 |
| Softmax | | X | 1.70 |

*Table 4.* Speedup compared to the oneDNN C++ implementation

## 5.5. Case study: Accelerating an operation of Llama 3

A common use case for kernel optimization is accelerating LMs on given hardware, often by allowing reduced precision as long as model performance is unaffected. As a case study, we target an operation in the Llama 3.2 1B model, the rotary positional embedding function, which presents a bottleneck on Intel hardware. We define a custom task (see Appendix C) with the PyTorch implementation (*apply_rotary_pos_emb*, combining unsqueeze + rotate-half) as the reference. Correctness is tested by comparing our kernel's output to the reference output, and by verifying that a full Llama3 model pass with our kernel yields identical results on a simple query. KernelFoundry discovers a correct kernel in just two iterations and achieves a 7.9× speedup within ten iterations. This accelerates the rotary embedding operation, resulting in an 8% reduction in total forward pass time (from 0.413 s to 0.38 s) on a B580 Intel GPU.

## 6. Conclusion

We presented KernelFoundry, an evolutionary framework for GPU kernel optimization with LLMs. We combine kernel-specific quality-diversity search, meta-prompting and parameter optimization to generate high-performing CUDA and SYCL kernels, demonstrating cross-platform applicability and outperforming SOTA methods. Our experiments highlight the need for open, reproducible benchmarks and kernel databases to advance the field.

Future work includes templated kernel generation for speedups across input ranges and tensor shapes, improved formal verification to eliminate reward hacking, and deeper hardware specialization. Despite the success of prompting-based frameworks, it remains a promising avenue to finetune models, in particular through RL with verifiable rewards. As LLMs and hardware evolve, frameworks like KernelFoundry will be essential for automated, robust kernel optimization – enabling rapid deployment of new language models and fast adaptation of kernels to new hardware platforms.

## Impact Statement

This work advances Machine Learning and High-Performance Computing through automated kernel optimization. By enabling efficient cross-platform GPU kernel generation, our work aims to democratize access to high-performance computing across diverse hardware ecosystems and improve sustainability through optimized energy efficiency. While automated code generation raises questions about code quality and security, our framework includes rigorous correctness validation to mitigate these concerns. We do not foresee specific negative societal consequences that must be highlighted beyond those common to advances in AI-assisted programming.

## Data and code availability

Code is available at https://github.com/isl-org/kernelfoundry.

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

# A. Summary of the Related Works

*Table 5.* Overview of related work on LLM-based kernel generation and optimization. Methods are grouped by approach: benchmarks, prompting-validation loops, multi-agent, and finetuning. Hardware is GPU unless denoted otherwise.

| Reference | Language | Hardware | Benchmark [Count] | Method | Key Features |
|---|---|---|---|---|---|
| *Benchmarks* | | | | | |
| KernelBench (Ouyang et al., 2025b) | CUDA / Triton | NVIDIA | KernelBench [250] | Repeated prompting | De-facto standard |
| MultiKernelBench (Wen et al., 2025) | CUDA / Pallas / AscendC | NVIDIA / TPU / NPU | KernelBench revised [285] | Prompting | Multi-platform |
| TritonBench (Li et al., 2025b) | Triton | NVIDIA | TritonBench [184] | Prompting | Triton-specific |
| NPUEval (Kalade & Schelle, 2025) | C++ AIE | AMD NPU | Custom [102] | Prompting + RAG | NPU focus |
| *Approaches based on inference-validation loops* | | | | | |
| Robust-kbench (Lange et al., 2025b) | CUDA | NVIDIA | KernelBench + Custom [269] | Evolve | Robustness focus |
| cuPilot (Chen et al., 2025a) | CUDA | NVIDIA | KernelBench L1 [100] | Evolve | Strategy coordination |
| KernelFalcon (Wang, 2025) | Triton | NVIDIA | KernelBench [250] | Repeated prompting | 100% correctness |
| Opal (Zaeed et al., 2025) | CUDA / HIP | NVIDIA / AMD | Custom [8] | Prompting + 3 profilers | Modular profiling |
| SwizzlePerf (Tschand et al., 2025) | HIP | AMD | Custom [10] | HW-aware prompting | Microarch knowledge |
| TritonForge (Li et al., 2025a) | Triton | NVIDIA | TritonBench [185] | Profiling-guided | Automated profiling |
| KernelEvolve (Liao et al., 2025) | Triton / CuTE | NVIDIA / AMD | KernelBench [250] | Tree search | Meta scaling |
| Geak (Wang et al., 2025) | Triton | AMD | TritonBench-revised | Prompting + Reflection | AI agent |
| PEAK (Tariq et al., 2025) | CUDA / HIP / HLSL | NVIDIA / AMD / Qualcomm | MatMul | Natural transformations | Iterative + Modular |
| *Multi-Agent Approaches* | | | | | |
| Astra (Wei et al., 2025) | CUDA | NVIDIA | Custom [3] | Multi-agent | Plan/Code/Test/Profile |
| CudaForge (Zhang et al., 2025) | CUDA | NVIDIA | KernelBench [25] | Coder + Judge | Hardware feedback |
| STARK (Dong et al., 2025) | CUDA | NVIDIA | KernelBench [250] | Plan + Code + Debug | Strategic agents |
| QiMeng (Zhu et al., 2026) | Triton | NVIDIA | KernelBench + TritonBench | Macro/Micro coding | Two-stage thinking |
| PRAGMA (Lei et al., 2025) | Triton / C++ | NVIDIA / Intel CPU | KernelBench | Coder + Verifier + Conductor | Profiling-reasoned |
| AKG Agent (Du et al., 2025) | Triton | NVIDIA / Ascend NPU | KernelBench L1 [100] | Coder + Designer | Cross-platform |
| *Finetuning Approaches* | | | | | |
| Kevin (Baronio et al., 2025) | CUDA | NVIDIA | KernelBench [100] | Multi-turn RL | Turn-level rewards |
| CUDA-L1 (Li et al., 2025c) | CUDA | NVIDIA | KernelBench [250] | Contrastive RL | 3.12× speedup |
| CUDA-L2 (Su et al., 2025) | CUDA | NVIDIA (A100) | HGEMM [1000] | Multi-stage RL + NCU | Beats cuBLAS (+11%) |
| TritonRL (Woo et al., 2025) | Triton | NVIDIA | KernelBench [200] | SFT + GRPO | No reward hacking |
| **KernelFoundry (Ours)** | **SYCL / CUDA / Triton** | **Intel / NVIDIA** | **KernelBench + Custom** | **MAP-Elites + Meta-prompt** | **QD + Parameter optim.** |

# B. Implementation Details

## B.1. System Configuration

KernelFoundry is implemented in Python with the following core dependencies:

- **LLM inference**: OpenAI API, Anthropic API, vLLM for local models
- **Compilation**: Intel oneAPI DPC++ 2025.2, NVIDIA CUDA Toolkit 12.8, Triton 3.5
- **Profiling**: Profiling Tools Interfaces for GPU (unitrace 2.3) for SYCL, NVIDIA Nsight Compute 2025.3.0.0 for CUDA
- **Testing**: PyTorch 2.9

## B.2. Benchmarking kernel runtime

In contrast to prior work, we implement several improvements in the kernel benchmarking to reduce the variance of runtime measurements. Our implementation proceeds as follows: First, we run a fixed number of initial trials to determine the rough runtime of the kernel. This initial measurement informs the number of warmup trials and main trials, which are set based on a minimal total *time* rather than a fixed amount of trials (if the kernel is slow, less warmup trials and main trials are required). Furthermore, we noticed that for very fast kernels, the `torch.cuda.synchronize` or `torch.xpu.synchronize` operation has a significant overhead. We reduce this overhead by running an inner loop within the main trials, such that multiple trials are executed before each synchronize.

For all experiments, we set the minimum warmup time to 1 second, the minimum warmup iterations to 10, the inner-loop minimum time to 0.01 seconds, the minimum number of main iterations to 10 and the minimum runtime of the main performance measurement to 1s. This ensures that measured durations are comfortably larger than the timer precision and synchronization overhead while guaranteeing enough measurements to compute runtime statistics.

Furthermore, we noticed that the runtime measurement of *backward* operations in robust-kbench has a significant overhead due to using `torch.autograd`, rather than measuring the kernel execution time directly. We found a way to measure the isolated runtime of the kernel which is more stable. However, for the reference we still need to measure

`torch.autograd.grad`. This leads to high speedups for backward operations as can be seen in Table 7; however, this appeared as a minor drawback to us compared to an incorrect kernel runtime measurement.

### B.3. Profiler Feedback

For correct kernels, optional profiling provides performance insights. We collect and provide the following information:

- **Execution time**: Wall-clock runtime
- **Memory bandwidth**: Achieved vs. theoretical peak (via `unitrace` or Nsight Compute)
- **Compute utilization**: ALU occupancy, instruction throughput
- **Bottleneck identification**: Memory-bound vs. compute-bound classification

Profiler feedback is structured into natural language summaries (e.g., "Kernel is memory-bound at 45% of peak bandwidth. Consider shared memory tiling to improve data reuse.").

### B.4. Hyperparameters

Table 6 lists the default hyperparameters used in our experiments. The number of iterations depends on the experiment: For a fair comparison to The AI CUDA Engineer (Lange et al., 2025a), we set the number of iterations to 40 and the population size to 4 to align with their experimental setup. Robust-kbench (Lange et al., 2025b) sample 8 suggestions from the LLM in each iteration and show the results over 40 iterations, so we use the same (40 x 8) as used for SYCL kernel generation (Table 2). As explained in section 4, we also align the LLMs to the ones in the baseline papers. For the SYCL experiment, we chose to prompt a powerful language model (Claude Sonnet 4.5) in the first iteration to avoid getting trapped in a local minimum at the beginning. After the first iteration, we use an ensemble of GPT 5 mini and GPT 4.1 (equal weights). After the 40 iterations, we run 2 iterations of parameter optimization.

| Parameter | Value |
|---|---|
| *Evolution* | |
| Max generations | 40 (*) |
| Population per generation | 8 (*) |
| Selection strategy | Curiosity-driven |
| Archive dimensions | 4 |
| Bins per dimension | 4 |
| *Evaluation* | |
| Warmup iterations | 10 |
| Timing iterations | 100 |
| Target speedup | $2.0\times$ |
| *Meta-prompting* | |
| Prompt update frequency | Every 10 generations |
| Max prompt mutations | 3 per update |
| Prompt archive size | 16 |
| *LLM* | |
| Temperature | 0.3 |
| Max tokens | 8000 |
| Top-p | 1 |

*Table 6.* Hyperparameters for KernelFoundry. (*) number of operations, population size, and LLM model vary dependent on the experiment.

## C. Distributed System & Custom Task Input

**Infrastructure**  As shown in Figure 4, the KernelFoundry pipeline utilizes a lightweight main thread that interacts with four servers. The server processes can run on a single machine or on multiple remote machines in the same network.

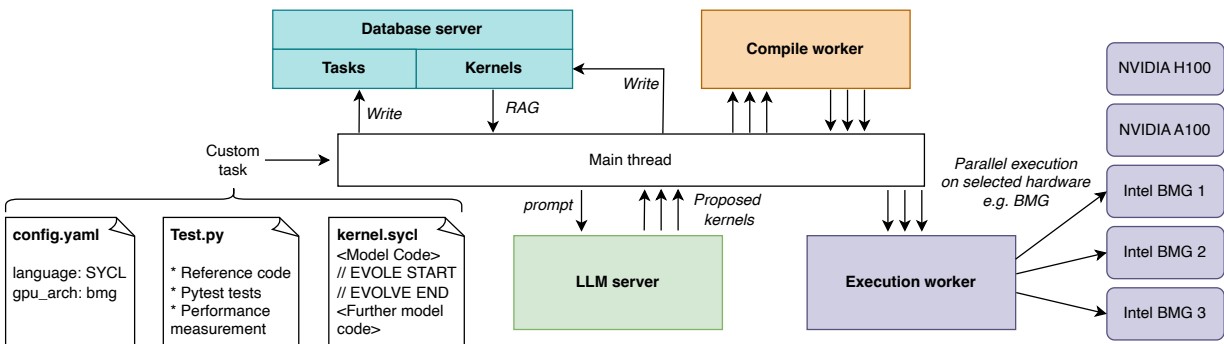

*Figure 4.* Overview of the KernelFoundry infrastructure and the custom task format

1. **LLM Server**: Hosts the generation model (Cloud services with REST-API or local vLLM instances).

2. **Compilation Workers**: Executes compiler toolchains (DPC++, nvcc). Does not need a GPU.

3. **Execution Workers**: Run correctness tests and benchmarks on GPU nodes. Connected via a task queue ensuring single-task-per-GPU isolation.

4. **Database Server**: Stores all generated kernels, evaluation results, and evolutionary state for reproducibility and analysis.

The LLM server, compile worker, and execution worker are connected via a load balancer and queue system and allows to scale the number of workers and increase parallelism.

**Custom task**  Tasks are defined by a set of files with special markers, making them highly customizable and easy to integrate into existing projects. A task is defined by a config file in YAML format containing hyperparameters; a python module with a build function and correctness and performance tests defined in the pytest framework; and a language specific file for the generated code. Special markers are used to define sections for the reference code, optional user instructions, and optional initial kernel implementations passed to the model.

## D. KernelBench task filtering

We follow Lange et al. (2025b) and METR (2025) and filter out KernelBench tasks that are flawed. Lange et al. (2025b) propose 5 criteria: (1) small output value range, (2) small output StD, (3) small output StD across any axis, (4) small impact of inputs on the output, (5) baseline inefficiencies. For our experiments, we select the same task set as (Lange et al., 2025b) to ensure full consistency.

However, it is worth noting that we generally believe that criterion (3) is too strict, as we did not observe issues with the kernels generated for such tasks, and e.g. for outputs with shape 2 along one axis the StD can naturally be low. Tasks with baseline inefficiencies (5) do not necessarily need to be removed either, as the comparison is still fair. Lange et al. (2025b) used an LLM to identify these tasks which sometimes gave wrong responses on closer inspection. Filtering according to criteria (1), (2) and (4) would retain 223 tasks across level 1, 2 and 3.

The set of representative tasks (20 L1, 20 L2, see Section 4) was selected based on the following criteria:

1. Diversity: 20 tasks from level 1, 20 from level 2, covering all kind of operations (matmul, conv, conv transpose, activation functions, etc.)

2. Uncompromised: We only select tasks that comply with criteria (1)-(5) by Lange et al. (2025b), i.e. tasks that are not listed as "compromised tasks" in (Lange et al., 2025b) Appendix A.2 and A.3.

3. Availability of correct baseline kernels: For a fair comparison to (Lange et al., 2025a), we select only tasks for which they publish a kernel as part of their Huggingface dataset, and filtering only for the ones where their fastest kernel (per task) passes our correctness tests. Since our tests are more strict than the test used in (Lange et al., 2025a), 53 out of the 229 kernels failed our tests.

# E. Prompts

## E.1. Main prompt

You are a <language> programming expert specializing in GPU kernel optimization. Given a reference <reference-language> implementation, your objective is to create a performant kernel with identical functionality. The code you generate will be pasted into an existing project. Make sure to follow the existing code structure and function signatures. The <language> code you generate will be saved in < kernel.cpp / kernel.cu > and loaded using `torch.utils.cpp_extension.load()`:

` ``load(name=task_name, sources=[<kernel.cpp / kernel.cu>]) ``

Later, the function will be called via `load(name=task_name, ...).forward` and thoroughly tested.

**Examples:**
Here is an example of a correct <language> kernel for a given <reference-language> reference:
<example-reference-code>
<example-kernel-code>

**Reference code / Task:**
This is the reference <reference-language> implementation:
<reference-code>

**Top performing kernel:**
This is the best kernel tested so far (Runtime: <top-kernel-runtime>):
<top-kernel-code>

**Last tested kernel:**
Here is the last kernel we tested (Runtime: <last-kernel-runtime>):
<last-kernel-code>
Console output from running this kernel:
<last-kernel-log>

**Hardware specification:**
Your code will run on the following hardware:
<hardware-specs>
Please consider the hardware specifications when improving the code.

**Main Instructions:**

1. Provide a functional kernel that matches the reference implementation.
2. Use constructs to efficiently run the code on GPU.
3. Provide the complete code in a code block.

**Optimization strategies:**
<evolvable-optimization-strategies>

**Critical Requirements:**

1. The kernel must exactly match the reference's functionality.
2. The code must compile and run properly on the GPU.
3. Do not cache or reuse previous results; ensure the code executes fully on each run.

**Response Format:**
Please structure your response as follows:

1. Analysis: Summarize the issues found in the previous kernel and log. Explain your proposed changes and optimizations.
2. Code: Provide the complete, improved <language> code in code blocks:

```
Your code here
```

### E.2. Templated kernels for parameter optimization

To guide the model towards converting parameters to template parameters which can then be tuned with our pipeline, we use the following prompt and code example:

> You are a <language> programming expert specializing in GPU kernel optimization. Your task is to optimize a given <language> kernel.
> **Given kernel:**
> Here is the <language> kernel that we tested:
> <Code of best generated kernel>
> To optimize this kernel for specific hardware, please propose a templated kernel with some template parameters that can be tuned. To do so, you need to write an extension for PyTorch that implements a templated <language> kernel and a forward function for dispatching. Select suitable parameter options by adding them as dispatch-options in the forward function.
>
> **Requirements:**
>
> - The kernel should be templated on suitable parameters, e.g., block size, etc.
>
> - The `forward_templated` function should launch the kernel with the given parameter values.
>
> - The `forward` function should have standard arguments corresponding to the template parameters of `forward_templated`, and should select the correct instantiation based on the input values.
>
> - The code must match the given kernel in functionality.
>
> - The code should include a Pybind11 interface exposing `forward`.
>
> Here is an example of a templated kernel in the correct format:

*Listing 1.* Example SYCL kernel with parameter dispatch

```
1  #include <sycl/sycl.hpp>
2  #include <torch/extension.h>
3  #include <c10/xpu/XPUStream.h>
4
5  // Kernel name struct at namespace scope
6  template <int BX, int BY>
7  struct ElementwiseMulKernel {};
8
9  template <int BLOCK_X, int BLOCK_Y>
10 void elementwise_mul_sycl_kernel(
11     torch::Tensor A,
12     torch::Tensor B,
13     torch::Tensor C,
14     int N,
15     int M
16 ) {
17     auto a_data = A.data_ptr<float>();
18     auto b_data = B.data_ptr<float>();
19     auto c_data = C.data_ptr<float>();
20
21     sycl::queue& q = c10::xpu::getCurrentXPUStream().queue();
22
23     sycl::range<2> global_range(
24         ((N + BLOCK_Y - 1) / BLOCK_Y) * BLOCK_Y,
25         ((M + BLOCK_X - 1) / BLOCK_X) * BLOCK_X
26     );
27     sycl::range<2> local_range(BLOCK_Y, BLOCK_X);
28
29     q.submit([&](sycl::handler& cgh) {
30         cgh.parallel_for<ElementwiseMulKernel<BLOCK_X, BLOCK_Y>>(
```

```
31              sycl::nd_range<2>(global_range, local_range),
32              [=](sycl::nd_item<2> item) {
33                  int row = item.get_global_id(0);
34                  int col = item.get_global_id(1);
35                  if (row < N && col < M) {
36                      int idx = row * M + col;
37                      c_data[idx] = a_data[idx] * b_data[idx];
38                  }
39              }
40          );
41      }).wait();
42  }
43
44  // 2. Templated forward function
45  template <int BLOCK_X, int BLOCK_Y>
46  torch::Tensor forward_templated(torch::Tensor A, torch::Tensor B) {
47      int N = A.size(0);
48      int M = A.size(1);
49      auto C = torch::empty({N, M}, A.options());
50      elementwise_mul_sycl_kernel<BLOCK_X, BLOCK_Y>(A, B, C, N, M);
51      return C;
52  }
53
54  // 3. Dispatcher - must have arguments corresponding to the template parameters of
        forward_templated
55  torch::Tensor forward(torch::Tensor A, torch::Tensor B, int block_x, int block_y) {
56      TORCH_CHECK(A.scalar_type() == torch::kFloat, "Only float32 supported in this example"
            );
57      TORCH_CHECK(A.dim() == 2 && B.dim() == 2, "Only 2D tensors supported");
58      TORCH_CHECK(A.sizes() == B.sizes(), "Input sizes must match");
59
60      if (block_x == 16 && block_y == 16) {
61          return forward_templated<16, 16>(A, B);
62      } else if (block_x == 32 && block_y == 8) {
63          return forward_templated<32, 8>(A, B);
64      } else if (block_x == 8 && block_y == 32) {
65          return forward_templated<8, 32>(A, B);
66      } else {
67          TORCH_CHECK(false, "Unsupported block size combination");
68      }
69  }
70
71  // 4. Pybind11 interface
72  PYBIND11_MODULE(TORCH_EXTENSION_NAME, m) {
73      m.def("forward", &forward, "Elementwise multiplication with block size dispatch");
74  }
```

### E.3. Example of optimization-aware prompting

To make optimization-aware prompting more tangible, consider the following example: A parent kernel achieved a runtime of 0.021 ms. The framework identified a positive fitness gradient along the memory dimension and injected the following natural-language hint into the next generation prompt: "Use register tiling: each thread computes an M×N output block in registers. Apply #pragma unroll to keep intermediate values in registers and avoid spilling. Combine with shared memory tiling for multi-level memory hierarchy optimization." The child kernel acted on this hint by replacing a flat accumulator array with an explicit 2×2 register tile, restructuring the inner loop to accumulate all four overlapping convolution outputs simultaneously, and adding #pragma unroll across all inner loops. This yielded a runtime of 0.0195 ms, a  7% improvement.

## F. Per-task results

### F.1. Robust-kbench comparison

| Operation | Robust-kbench original | Robust-kbench re-evaluated | Ours | Ours + parameter optim. |
|---|---|---|---|---|
| layernorm_forward | 165.678 | 55.116 | 55.625 | 55.625 |
| llama_ffw | 1.011 | 1.010 | 1.014 | 1.012 |
| llama_rmsnorm_forward | 3.920 | 2.911 | 3.529 | 3.517 |
| mnist_conv_relu_pool_forward | 2.790 | 3.947 | 5.068 | 5.007 |
| mnist_cross_entropy_backward | 0.958 | 10.388 | 23.905 | 23.905 |
| mnist_cross_entropy_forward | 1.933 | 0.400 | 1.434 | 1.389 |
| mnist_linear_backward | 1.200 | 9.700 | 22.000 | 22.000 |
| mnist_linear_forward | 2.517 | 1.558 | 1.669 | 1.520 |
| mnist_linear_relu_backward | 1.226 | 15.446 | 22.233 | 22.233 |
| mnist_linear_relu_forward | 2.494 | 1.345 | 2.726 | 2.675 |
| mnist_pool_backward | 1.116 | 3.354 | 4.761 | 4.761 |
| resnet_block | 2.625 | 1.205 | 1.325 | 1.328 |
| Average | 15.622 | 8.865 | 12.107 | 12.081 |

*Table 7.* Comparison on robust-kbench by task

## F.2. Comparison to AI CUDA Engineer

| Operation | Level | AI CUDA Engineer original | AI CUDA Engineer re-evaluated | Ours | Ours + parameter optim. |
|---|---|---|---|---|---|
| 20_LeakyReLU | 1 | 1.134 | 0.621 | 0.910 | 0.955 |
| 21_Sigmoid | 1 | 1.109 | 0.554 | 0.805 | 0.805 |
| 25_Swish | 1 | 1.562 | 1.016 | 1.349 | 1.402 |
| 30_Softsign | 1 | 2.474 | 1.724 | 2.895 | 2.895 |
| 33_BatchNorm | 1 | 0.619 | 0.975 | 1.017 | 1.017 |
| 44_Average_Pooling_1D | 1 | 1.238 | 0.643 | 0.769 | 0.824 |
| 48_Mean_reduction_over_a_dimension | 1 | 1.760 | 0.567 | 0.887 | 0.887 |
| 4_Matrix_vector_multiplication_ | 1 | 0.939 | 0.897 | 0.985 | 0.990 |
| 53_Min_reduction_over_a_dimension | 1 | 2.187 | 0.981 | 1.317 | 1.317 |
| 5_Matrix_scalar_multiplication | 1 | 0.999 | 0.999 | 1.003 | 1.008 |
| 64_conv_transposed_1D | 1 | 1.021 | 0.578 | 1.017 | 1.213 |
| 67_conv_standard_1D | 1 | 1.411 | 1.199 | 1.566 | 1.576 |
| 72_ConvTranspose3d_BatchNorm_AvgPool_AvgPool | 1 | 1.047 | 1.030 | 0.315 | 0.319 |
| 76_conv_standard_1D_dilated_strided__ | 1 | 1.883 | 1.209 | 1.325 | 1.325 |
| 7_Matmul_with_small_K_dimension_ | 1 | 0.670 | 0.612 | 1.095 | 1.135 |
| 82_conv_depthwise_2D_square_input_square_kernel | 1 | 1.239 | 1.530 | 0.794 | 0.794 |
| 86_conv_depthwise_separable_2D | 1 | 0.808 | 0.574 | 0.940 | 0.961 |
| 87_conv_pointwise_2D | 1 | 0.249 | 0.499 | 1.000 | 1.000 |
| 89_cumsum | 1 | 2.214 | 1.319 | 1.824 | 2.127 |
| 99_TripletMarginLoss | 1 | 3.873 | 2.570 | 2.278 | 2.278 |
| Average | 1 | 1.422 | 1.005 | 1.204 | 1.241 |
| 16_ConvTranspose2d_Mish_Add_Hardtanh_Scaling | 2 | 1.689 | 1.684 | 0.466 | 0.469 |
| 17_Conv2d_InstanceNorm_Divide | 2 | 1.786 | 2.036 | 1.920 | 2.022 |
| 1_Conv2D_ReLU_BiasAdd | 2 | 1.170 | 1.626 | 3.322 | 3.444 |
| 21_Conv2d_Add_Scale_Sigmoid_GroupNorm | 2 | 1.786 | 1.982 | 2.891 | 2.834 |
| 24_Conv3d_Min_Softmax | 2 | 1.001 | 1.026 | 1.013 | 1.013 |
| 32_Conv2d_Scaling_Min | 2 | 1.767 | 1.537 | 2.398 | 2.655 |
| 35_Conv2d_Subtract_HardSwish_MaxPool_Mish | 2 | 1.932 | 2.331 | 1.624 | 1.669 |
| 37_Matmul_Swish_Sum_GroupNorm | 2 | 1.499 | 1.804 | 1.256 | 1.443 |
| 46_Conv2d_Subtract_Tanh_Subtract_AvgPool | 2 | 1.393 | 2.333 | 5.560 | 5.622 |
| 47_Conv3d_Mish_Tanh | 2 | 1.086 | 1.192 | 1.361 | 1.379 |
| 50_ConvTranspose3d_Scaling_AvgPool_BiasAdd_Scaling | 2 | 1.000 | 1.019 | 3.039 | 3.099 |
| 59_Matmul_Swish_Scaling | 2 | 1.911 | 0.954 | 1.412 | 1.408 |
| 5_ConvTranspose2d_Subtract_Tanh | 2 | 1.092 | 1.153 | 0.199 | 0.200 |
| 67_Conv2d_GELU_GlobalAvgPool | 2 | 1.634 | 2.000 | 1.464 | 1.526 |
| 70_Gemm_Sigmoid_Scaling_ResidualAdd | 2 | 1.713 | 0.781 | 1.592 | 1.881 |
| 73_Conv2d_BatchNorm_Scaling | 2 | 1.024 | 0.924 | 1.739 | 1.741 |
| 82_Conv2d_Tanh_Scaling_BiasAdd_Max | 2 | 1.999 | 2.712 | 3.883 | 3.744 |
| 85_Conv2d_GroupNorm_Scale_MaxPool_Clamp | 2 | 1.241 | 1.322 | 2.037 | 2.077 |
| 97_Matmul_BatchNorm_BiasAdd_Divide_Swish | 2 | 2.277 | 1.688 | 1.662 | 1.726 |
| 99_Matmul_GELU_Softmax | 2 | 2.778 | 2.020 | 2.183 | 2.124 |
| Average | 2 | 1.589 | 1.606 | 2.051 | 2.104 |

*Table 8.* Comparison to the AI CUDA Engineer (Lange et al., 2025a) by KernelBench task

## F.3. Comparison to OpenEvolve

| Operation | Ours | OpenEvolve |
|---|---|---|
| 16_ConvTranspose2d_Mish_Add_Hardtanh_Scaling | 1.786 | 1.792 |
| 17_Conv2d_InstanceNorm_Divide | 3.550 | 1.851 |
| 1_Conv2D_ReLU_BiasAdd | 2.917 | 3.295 |
| 21_Conv2d_Add_Scale_Sigmoid_GroupNorm | 3.045 | 6.612 |
| 24_Conv3d_Min_Softmax | 1.207 | 1.211 |
| 32_Conv2d_Scaling_Min | 7.871 | 10.296 |
| 35_Conv2d_Subtract_HardSwish_MaxPool_Mish | 1.972 | 1.985 |
| 37_Matmul_Swish_Sum_GroupNorm | 1.928 | 0.520 |
| 46_Conv2d_Subtract_Tanh_Subtract_AvgPool | 5.005 | 3.446 |
| 47_Conv3d_Mish_Tanh | 0.834 | 0.747 |
| 50_ConvTranspose3d_Scaling_AvgPool_BiasAdd_Scaling | 1.667 | 1.684 |
| 59_Matmul_Swish_Scaling | 0.913 | 0.674 |
| 5_ConvTranspose2d_Subtract_Tanh | 1.262 | 1.237 |
| 67_Conv2d_GELU_GlobalAvgPool | 2.498 | 4.061 |
| 70_Gemm_Sigmoid_Scaling_ResidualAdd | 0.615 | 0.767 |
| 73_Conv2d_BatchNorm_Scaling | 0.904 | 0.748 |
| 82_Conv2d_Tanh_Scaling_BiasAdd_Max | 7.955 | 2.023 |
| 85_Conv2d_GroupNorm_Scale_MaxPool_Clamp | 3.031 | 3.043 |
| 97_Matmul_BatchNorm_BiasAdd_Divide_Swish | 1.441 | 0.471 |
| 99_Matmul_GELU_Softmax | 4.234 | 4.239 |
| Average | 2.732 | 2.535 |

*Table 9.* Comparison to OpenEvolve (Sharma, 2025) by KernelBench task

## F.4. Ablation study with respect to prompt evolution and gradient guidance

| Operation | W/o prompt evolution | Prompt evolution (every 10) | Prompt evolution (every 2) | W/o gradient |
|---|---|---|---|---|
| 16_ConvTranspose2d_Mish_Add_Hardtanh_Scaling | 0.294 | 1.638 | 1.654 | 1.633 |
| 17_Conv2d_InstanceNorm_Divide | 3.249 | 3.293 | 1.636 | 2.866 |
| 1_Conv2D_ReLU_BiasAdd | 2.500 | 2.015 | 3.296 | 1.968 |
| 21_Conv2d_Add_Scale_Sigmoid_GroupNorm | 3.169 | 2.807 | 2.810 | 2.144 |
| 24_Conv3d_Min_Softmax | 1.504 | 1.089 | 1.098 | 1.103 |
| 32_Conv2d_Scaling_Min | 9.541 | 9.196 | 6.446 | 8.455 |
| 35_Conv2d_Subtract_HardSwish_MaxPool_Mish | 2.759 | 6.831 | 8.717 | 1.723 |
| 37_Matmul_Swish_Sum_GroupNorm | 1.452 | 0.775 | 1.727 | 1.983 |
| 46_Conv2d_Subtract_Tanh_Subtract_AvgPool | 4.747 | 3.592 | 1.774 | 1.728 |
| 47_Conv3d_Mish_Tanh | 1.368 | 0.649 | 0.525 | 0.646 |
| 50_ConvTranspose3d_Scaling_AvgPool_BiasAdd_Scaling | 2.617 | 1.221 | 2.972 | 1.565 |
| 59_Matmul_Swish_Scaling | 0.653 | 0.656 | 0.831 | 0.647 |
| 5_ConvTranspose2d_Subtract_Tanh | 0.111 | 0.729 | 1.176 | 0.617 |
| 67_Conv2d_GELU_GlobalAvgPool | 3.247 | 3.868 | 2.591 | 2.971 |
| 70_Gemm_Sigmoid_Scaling_ResidualAdd | 0.328 | 0.976 | 0.692 | 0.660 |
| 73_Conv2d_BatchNorm_Scaling | 0.696 | 1.670 | 2.057 | 1.698 |
| 82_Conv2d_Tanh_Scaling_BiasAdd_Max | 3.661 | 3.844 | 4.760 | 4.175 |
| 85_Conv2d_GroupNorm_Scale_MaxPool_Clamp | 2.159 | 2.809 | 2.674 | 3.142 |
| 97_Matmul_BatchNorm_BiasAdd_Divide_Swish | 0.723 | 0.462 | 0.627 | 0.623 |
| 99_Matmul_GELU_Softmax | 5.324 | 2.657 | 2.787 | 3.208 |
| Average | 2.505 | 2.539 | 2.542 | 2.178 |
| Geometric mean | 1.600 | 1.833 | 1.944 | 1.695 |

*Table 10.* Ablation studies with respect to prompt evolution and gradient guidance.

## F.5. Evaluation of hardware-awareness

| Operation | Runtime [ms] on LNL | | | Runtime [ms] on B580 | | |
|---|---|---|---|---|---|---|
| | Optimized on LNL | Optimized on B580 | hws | Optimized on LNL | Optimized on B580 | hws |
| 16_ConvTranspose2d_Mish_Add_Hardtanh_Scaling | 3.130 | 3.140 | 1.003 | 0.850 | 0.866 | 0.982 |
| 17_Conv2d_InstanceNorm_Divide | 0.269 | 0.401 | 1.491 | 0.090 | 0.071 | 1.268 |
| 1_Conv2D_ReLU_BiasAdd | 0.253 | 0.214 | 0.846 | 0.055 | 0.025 | 2.228 |
| 21_Conv2d_Add_Scale_Sigmoid_GroupNorm | 0.532 | 0.525 | 0.987 | 0.073 | 0.050 | 1.474 |
| 24_Conv3d_Min_Softmax | 6.130 | 6.150 | 1.003 | 1.820 | 1.840 | 0.989 |
| 32_Conv2d_Scaling_Min | 0.088 | 0.454 | 5.142 | 0.041 | 0.040 | 1.010 |
| 35_Conv2d_Subtract_HardSwish_MaxPool_Mish | 0.465 | 0.476 | 1.024 | 0.053 | 0.048 | 1.109 |
| 37_Matmul_Swish_Sum_GroupNorm | 0.075 | 0.110 | 1.473 | 0.033 | 0.055 | 0.608 |
| 46_Conv2d_Subtract_Tanh_Subtract_AvgPool | 0.199 | 0.527 | 2.648 | 0.076 | 0.071 | 1.075 |
| 47_Conv3d_Mish_Tanh | 1.630 | 1.160 | 0.712 | 0.420 | 0.263 | 1.597 |
| 50_ConvTranspose3d_Scaling_AvgPool_BiasAdd_Scaling | 50.500 | 36.700 | 0.727 | 16.100 | 10.400 | 1.548 |
| 59_Matmul_Swish_Scaling | 0.127 | 0.128 | 1.008 | 0.051 | 0.044 | 1.155 |
| 5_ConvTranspose2d_Subtract_Tanh | 0.959 | 0.977 | 1.019 | 0.199 | 0.196 | 1.015 |
| 67_Conv2d_GELU_GlobalAvgPool | 0.265 | 0.211 | 0.796 | 0.090 | 0.085 | 1.056 |
| 70_Gemm_Sigmoid_Scaling_ResidualAdd | 0.182 | 0.180 | 0.989 | 0.056 | 0.055 | 1.013 |
| 73_Conv2d_BatchNorm_Scaling | 0.960 | 1.200 | 1.250 | 0.537 | 0.473 | 1.135 |
| 82_Conv2d_Tanh_Scaling_BiasAdd_Max | 0.132 | 0.530 | 4.015 | 0.048 | 0.094 | 0.510 |
| 85_Conv2d_GroupNorm_Scale_MaxPool_Clamp | 0.485 | 0.652 | 1.344 | 0.055 | 0.141 | 0.391 |
| 97_Matmul_BatchNorm_BiasAdd_Divide_Swish | 0.102 | 0.126 | 1.235 | 0.055 | 0.057 | 0.974 |
| 99_Matmul_GELU_Softmax | 0.009 | 0.018 | 2.027 | 0.009 | 0.009 | 1.038 |
| Average | 3.325 | 2.694 | 1.537 | 1.036 | 0.744 | 1.109 |

*Table 11.* Crossover-experiment: Kernels are optimized in two separate runs on an Intel B580 and Intel LNl GPU, and then benchmarked on the respective other hardware as well. The hardware-speedup $hws$ is the ratio of both runtimes. In most cases, the kernel optimized for the target GPU outperforms the kernel that was optimized on the other GPU (lower runtime on the target GPU, $hws > 1$)

# G. Results for open-source model (GPT-OSS 20B)

For the sake of reproducibility, we provide results of running KernelFoundry with GPT-OSS 20B in Table 12. We use the representative subset of 20 kernels of KernelBench level 2 and prompt the model to generate SYCL kernels. Unfortunately, the model's lower capabilities led to failure in generating correct kernels in 7 out of 20 cases, even after 40 iterations and using a population of 4 per generation. However, in the successful cases, the kernels were optimized, achieving, for instance, a speedup of 1.775 for operation 16 on an Intel LNL GPU.

| Operation | Speedup |
|---|---|
| 16_ConvTranspose2d_Mish_Add_Hardtanh_Scaling.py | 1.775 |
| 17_Conv2d_InstanceNorm_Divide.py | - |
| 1_Conv2D_ReLU_BiasAdd.py | 1.708 |
| 21_Conv2d_Add_Scale_Sigmoid_GroupNorm.py | - |
| 24_Conv3d_Min_Softmax.py | 0.989 |
| 32_Conv2d_Scaling_Min.py | 2.106 |
| 35_Conv2d_Subtract_HardSwish_MaxPool_Mish.py | 0.865 |
| 37_Matmul_Swish_Sum_GroupNorm.py | 0.043 |
| 46_Conv2d_Subtract_Tanh_Subtract_AvgPool.py | 1.576 |
| 47_Conv3d_Mish_Tanh.py | 0.304 |
| 50_ConvTranspose3d_Scaling_AvgPool_BiasAdd_Scaling.py | - |
| 5_ConvTranspose2d_Subtract_Tanh.py | 1.223 |
| 59_Matmul_Swish_Scaling.py | - |
| 67_Conv2d_GELU_GlobalAvgPool.py | 0.714 |
| 70_Gemm_Sigmoid_Scaling_ResidualAdd.py | - |
| 73_Conv2d_BatchNorm_Scaling.py | 0.031 |
| 82_Conv2d_Tanh_Scaling_BiasAdd_Max.py | 1.085 |
| 85_Conv2d_GroupNorm_Scale_MaxPool_Clamp.py | - |
| 97_Matmul_BatchNorm_BiasAdd_Divide_Swish.py | 0.069 |
| 99_Matmul_GELU_Softmax.py | - |

*Table 12.* Results for GPT-OSS for reproducibility.

## H. Convergence curves

Figure 5 shows convergence over trials corresponding to the heatmap in Figure 3, but reports the best speedup per iteration rather than the cumulative best. This reveals that some iterations are more exploratory, yielding lower speedups as the system searches broader regions of the solution space. Figure 6 presents two examples of convergence behavior when scaling to 100 iterations. Even after 80 iterations, KernelFoundry continues to discover higher-performing kernels, particularly when using meta-prompting.

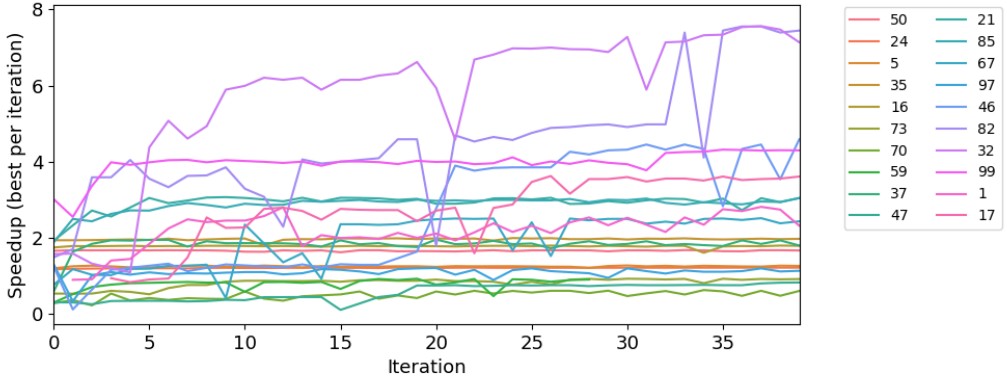

*Figure 5.* Speedup per iteration

## I. Cost estimation

We measured token usage for five KernelBench level 2 problems using the same experimental configuration as Table 1 (40 iterations, 4 branches, o3-mini). Average token consumption per kernel was 11,102 input tokens and 5,377 output tokens. Using OpenAI's pricing model (o3-mini: $1.10 per 1M input tokens, $4.40 per 1M output tokens), the cost to generate 160 kernels (40 iterations × 4 branches) amounts to approximately $5.74 per kernel. With more recent models (gpt-5.4), costs increase to $17.34 per kernel. These costs are comparable to related work: robust-kbench reports $5 API cost per kernel, while METR demonstrates convergence with costs up to $20 per kernel for o3-mini. Higher costs are justified when tools substantially reduce manual engineering effort.

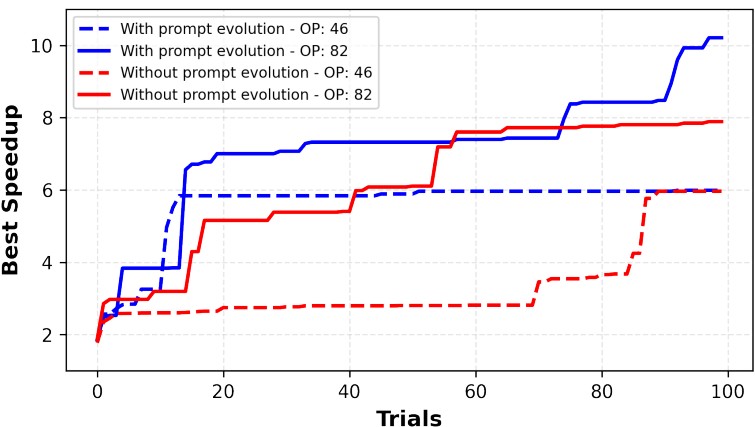

*Figure 6.* Convergence of the cumulative speedup over 100 iterations, shown for two examples, with and without prompt evolution activated.

## J. Qualitative analysis of prompt evolution

To better understand how prompts evolve during the training process, we analyzed the changes in prompts generated across training iterations. Figure 7 visualizes pairwise similarities between prompts generated at different training steps for randomly selected problems, computed using difflib's SequenceMatcher.

The analysis reveals distinct patterns in prompt evolution. While prompts change substantially for some problems, others exhibit clustering behavior where prompts remain similar across multiple iterations. Notably, we observe recurring patterns, with similarities appearing between prompts generated early in training and those generated substantially later, suggesting convergence toward stable prompt structures.

Manual inspection indicates that the "optimization strategies" and "common pitfalls" sections undergo the most frequent modifications, while "optimization philosophy" and "analysis guidance" remain relatively stable (see Section 3.5). Observed additions to prompts include specific anti-patterns and their fixes, such as "Reinterpreting host pointers to vector types inside kernels is unsafe unless the pointer is device-USM and properly aligned. Anti-pattern seen in generated code `reinterpret_castsycl::float4*(conv_out_ptr)[i]`. Fix: ...". Furthermore, additions include context-aware recommendations such as "Use direct convolution with shared memory when kernel sizes are small (3×3, 5×5) and memory reuse within a tile is high". These findings suggest that the training process effectively learns to refine domain-specific guidance based on problem characteristics.



*Figure 7.* Prompt similarity (of evolvable prompt regions) between iterations. Prompts are modified but patterns reoccur.

