# OpenReview forum: "KernelFoundry: Hardware-Aware Evolutionary GPU Kernel Optimization"
_ICML.cc/2026/Conference — ICML 2026 regular_

### Official Review · Reviewer_7Je4 · 2026-03-06

**Soundness:** 3
**Presentation:** 3
**Significance:** 4
**Originality:** 4
**Overall Recommendation:** 4
**Confidence:** 3

**Summary:**

This paper introduces KernelFoundry, a hardware-aware evolutionary framework for optimizing GPU kernels using LLMs. The framework integrates MAP-Elites quality-diversity search using kernel-specific behavioral descriptors, meta-prompt co-evolution to address context degradation, and a template-based parameter optimization for hardware-dependent kernel parameters. KernelFoundry targets both CUDA and SYCL, supporting multi-vendor backends and real-world tasks via a flexible input format and distributed execution infrastructure. The authors evaluate KernelFoundry on KernelBench, robust-kbench, and custom workloads, demonstrating consistent performance improvements over prior works and showcasing adaptability across multiple hardware types.

**Compliance With Llm Reviewing Policy:**

Affirmed.

**Key Questions For Authors:**

The citation for the method CodeEvolve is listed as "Team, S. A. CodeEvolve: Meta-prompting for evolutionary code generation, 2024. Meta-prompting for holistic prompt evolution in code generation." However, I cannot find this specific citation; I only find the paper "CodeEvolve: An open source evolutionary coding agent for algorithm discovery and optimization." Which citation is the correct one?

**Limitations:**

yes

**Strengths And Weaknesses:**

## Strengths

1. The paper combines MAP-Elites quality-diversity evolutionary search with kernel-specific behavioral descriptors, providing a structured and interpretable way to explore diverse GPU implementation strategies.

2. Meta-prompt evolution is used to co-evolve prompts alongside generated kernels, which is a creative mechanism to counteract context pollution and prompt drift.

3. The use of templated kernel generation and parameter optimization adds an additional layer of search that effectively decouples architectural variations from low-level parameter tuning.



## Weaknesses

1. Hardware generalization is demonstrated using two Intel GPUs (Arc 140V and Battlemage B580), but the diversity here is limited. Demonstrating cross-vendor portability (e.g., AMD and more Nvidia architectures) would further substantiate the hardware-agnostic claim.


2. While visualizations are generally strong, Figure 3’s heatmap could be supplemented by convergence curves or mode collapse diagnostics to provide deeper insight into evolutionary search dynamics.

3. The efficacy of meta-prompt evolution is shown in aggregate, but the paper could do more to disentangle its impact versus that of archive-based diversity or parameter search (no dedicated ablation or qualitative before/after prompt evolution examples).

---

> ### Author Rebuttal · Authors · 2026-03-30
>
> Thank you for the positive feedback. Additional results are provided in https://anonymous.4open.science/#!/r/kernelfoundry_rebuttal-79EE, see responses in the following.
>
> **[W1] Hardware-agnostic & cross-vendor portability**
>
> We claim that it is hardware-agnostic because cross-vendor portability is a main feature of SYCL, as was testified in related work, citing [1]: “While SYCL does not solve all the challenges of performance portability […] it does provide a single programming model and ecosystem to target most current HPC architectures productively”. To demonstrate this, we took a SYCL matrix multiplication kernel generated for Intel hardware and execute it on an NVIDIA platform. No code changes to the kernel are necessary, only compiler tooling changes. The SYCL kernel ran in 10.19 ms on an Intel Arc 140V GPU and in 2.92 ms on an NVIDIA GeForce RTX 3090. In the short time of the rebuttal, we unfortunately did not have access to AMD hardware.
>
> [1] Reguly, Istvan Z. "Evaluating the performance portability of SYCL across CPUs and GPUs on bandwidth-bound applications." Proceedings of the SC'23 Workshops of The International Conference on High Performance Computing, Network, Storage, and Analysis. 2023.
>
> **[W2] Convergence curves for heatmap**
>
> Thank you for this suggestion, we agree that convergence is difficult to assess from the heatmap alone. To address this, we provide two additional line plot figures (see Anonymous-GitHub link above):
> * Figure *speedup_40_cumulative*: corresponds to Figure 3 in the paper, showing the cumulative best speedup over iterations, making the overall optimization trajectory clearer.
> * Figure *speedup_40_per_iteration*: shows the best speedup achieved per iteration, revealing that some iterations are more exploratory in nature, producing lower speedups as the system searches broader regions of the solution space.
>
> Regarding mode collapse: while it is inherently difficult to quantify directly, the figure shows continued runtime variation in later iterations, suggesting the search does not degenerate into repetitive solutions. For a more comprehensive analysis, we refer to our response to Reviewer TdVw (W1) and Figure *normalized_100_trials*, which examines behavior over a longer 100-iteration run and provides further evidence against premature convergence.
>
> **[W3] No dedicated ablation study for prompt evolution**
>
> To directly address this concern, we ran a dedicated ablation experiment comparing three conditions: (1) no prompt evolution, (2) prompt evolution applied after every second kernel generation, and (3) prompt applied less frequently (SYCL optimization, 20 trials, population size=4). Results are provided in *table_ablations* at https://anonymous.4open.science/#!/r/kernelfoundry_rebuttal-79EE. Prompt evolution yields a moderate performance improvement, with more frequent application providing additional gains.
>
> We acknowledge that the speedup after 20 iterations may not fully show the benefit of prompt evolution, as it should mainly prevent context degradation after many iterations. Figure *prompt_evolution_100trials* shows a qualitative comparison over 100 iterations, demonstrating that (i) speedup increases faster when prompt evolution is enabled, and (ii) improvements continue to emerge late in the optimization run. In our response to Reviewer TdVw (Q2), we further analyze how the prompts evolve qualitatively across iterations. We will incorporate this ablation into the camera-ready version.
>
> **[Q1] Citation for CodeEvolve**
>
> Thank you for pointing out this issue – we originally only added a reference to the CodeEvolve GitHub which was displayed in a wrong way. The reference you found ("CodeEvolve: An open-source evolutionary coding agent for algorithm discovery and optimization.") is the correct one. We will correct it in the paper.

---

> > ### Author Rebuttal · Reviewer_7Je4 · 2026-04-06
> >
> > Thank you for the rebuttal. It has addressed most of my concerns.

---

### Official Review · Reviewer_XcbN · 2026-03-12

**Soundness:** 3
**Presentation:** 3
**Significance:** 3
**Originality:** 3
**Overall Recommendation:** 4
**Confidence:** 4

**Summary:**

GPU kernel optimization challenges LLMs is an interesting problem that introduces new challenges.
KernelFoundry paper proposes an evolutionary framework for hardware-aware optimization of GPU kernels. One of the key points seem to be that it addresses "mode collapse" (where models repeatedly propose similar, suboptimal variants) and "context degradation" (where failed attempts clutter the prompt context).

Key ideas of the paper are: (1) MAP-Elites Evolutionary Search where the framework maintains a diverse archive of high-performing kernels. It indexes these kernels along domain-specific behavioral dimensions; (2) Meta-Prompt Evolution where LLM analyzes generation failures and successes to refine the optimization guidance; (3) Template-Based Parameter Tuning where the system allows LLMs to generate "templated kernels" where hardware-dependent values; (4) Gradient-Informed Evolution where the framework tracks "fitness gradients" from parent-to-child transitions to provide the LLM with actionable natural-language mutation hints, such as "consider adding shared memory tiling".

In addition, instead of simply focusing on CUDA, KernelFoundry emphasizes SYCL, an open-standard C++ abstraction.  This seems to help support wide range of hardware. It seems to have a distributed architecture that allows rapid benchmarking and parallel evaluation across heterogeneous hardware.

**Compliance With Llm Reviewing Policy:**

Affirmed.

**Final Justification:**

The paper develops KernelFoundry which seems to be interesting way to generate performant kernels. My main concerns were on (A) the generality (possible extension to Triton), (B) limited scope (isolated kernel development vs end-to-end optimization), (C) balancing between LLM generation and templated kernels, and (D) regarding better presentation of mutation hints. All points were fully addressed by the author's rebuttal. I like the paper and I think it would be great inclusion to the program.

**Key Questions For Authors:**

I would like to stay positive about the paper given that this is an interesting problem and that the authors are proposing interesting tricks to prevent issues like "mode collapse".

* Effort to support wide range of hardware by using SYCL seems promising. What about Triton? Would that simplify the generation while helping the framework achieve good performance?
* Can you provide more evaluation on end-to-end inference? Sometimes isolated kernels or small scale experiments hide a lot of issues that only emanate in large-scale environments.
* Can you elaborate more on what would be the optimal balance between LLM generation vs using templated kernels for automated parameter tuning?
* Natural language mutation hints could be helpful in making this more interpretable. Can you provide a more detailed example in the paper such that the overall process becomes more relatable?

**Limitations:**

The authors seems to acknowledge the risk of "reward hacking"

**Strengths And Weaknesses:**

Strengths:
* Diverse Exploration (MAP-Elites) seems like an interesting way to prevent "mode collapse".
* Effort to support wide range of hardware by using SYCL seems promising.
* Hardware aware approach where it uses templated kernels for automated parameter tuning when beneficial.
* Natural language mutation hints could be helpful in making this more interpretable.

Weaknesses:
* The framework's distributed architecture and evolutionary loop require significant computational resources.
* There remains a risk of "reward hacking"
* While the paper showcases a case study on a Llama 3 operation, the primary focus is still on isolated kernel optimization.

---

> ### Author Rebuttal · Authors · 2026-03-30
>
> Thank you for the thoughtful review. We appreciate your support and address remaining concerns in the following.
>
> **[W1] Significant computational resources**
>
> First, note that non-LLM resources are not that large, especially because of our distributed infrastructure (see Figure 4 in the appendix) that ensures that only the actual testing is run on GPU, whereas all other parts (sampling the prompt, compilation, postprocessing) can be run on standard hardware. GPU hours are thus not high, and several jobs can be run in parallel.
> For a simple reproducible setup, there is the non-distributed version implemented that allows to use a local GPU.
> Secondly, LLM-related costs are comparable to related work and not high compared to the manual engineer effort involved with kernel generation. See Reviewer aQvs /W3 for a detailed cost analysis.
>
> **[W2] Risk of reward hacking**
>
> Reward hacking is a general problem in the field, and it is important to address. We have made several steps to prevent reward hacking, including 1) specific instructions in the prompt to forbid workarounds, 2) testing on a carefully selected subset of KernelBench (see Appendix D), and 3) following best practices from prior work that focuses specifically on robustness (robust-kbench). Kernels with surprisingly high speedup were inspected manually.
>
> **[W3] “Primary focus is on isolated kernel optimization”**
>
> We agree that isolated kernel optimization is not the same challenge as real-world tasks. A major engineering contribution of our work (but one that is hard to quantify) is our distributed pipeline with a customizable input format (see Appendix Figure 4) which enables more realistic kernel generation tasks. Since the paper submission, we have focused on running additional real use cases, including:
> * Generating an attention kernel For Llama 3.1 8B at 1 token decode stage achieving 98% of the performance of the highly optimized SDPA kernel in Pytorch.
> * Optimizing a SYCL kernel for CSR reduction by rows from the oneDAL library for which KernelFoundry found a kernel with 4.47 speedup within 10 iterations.
>
> We hope that these use cases provide further evidence on the flexibility of the framework. We will add these use cases to the appendix.
>
> **[Q1] Triton as an alternative**
>
> Triton is indeed supported on many platforms and is easy to use, but covers other use cases than SYCL (or CUDA) from our perspective. Most importantly, triton uses a JIT compilation process and auto-tiles, which is unsuitable for libraries (shipped binaries) where you need stable behaviour. Also SYCL's programming model provides more fine-grained control over the memory hierarchy and access to hardware features.
> Having said that, KernelFoundry is generic wrt the language and already supports Triton.
>
> **[Q2]** - See W3
>
> **[Q3] Optimal balance between LLM generation vs using templated kernels**
>
> How to mix algorithmic optimization vs parameter optimization is an interesting question. In initial experiments, we actually included the option for templated kernels in every prompt, hoping that the LLM would optimize the algorithm and parameters at the same time. However, we found that this increases the error rate and reduced the speedup:
>
> Experiment with 9 operations, 40 iterations, 8 branches, SYCL:
> * With templated option: 48% correct, avg speedup of 2.4
> * Without templated option: 52% correct, avg speedup of 2.9
>
> We concluded that it is difficult to combine these two parts, and that it is better to do algorithmic optimization first, and to do parameter optimization as a second step, as done in the experiments reported in the paper.
>
> **[Q4] More detailed example of natural language mutation hints**
>
> Thanks for the suggestion, we will add a full example to the paper. Here is a short version:
> A parent kernel achieved a runtime of 0.021ms. The framework identified a positive fitness gradient along the memory dimension and injected the following natural-language hint into the next generation prompt:
> "Use register tiling: each thread computes an M×N output block in registers. Apply #pragma unroll to keep intermediate values in registers and avoid spilling. Combine with shared memory tiling for multi-level memory hierarchy optimization."
> The child kernel acted on this hint by replacing a flat accumulator array with an explicit 2×2 register tile, restructuring the inner loop to accumulate all four overlapping convolution outputs simultaneously, and adding #pragma unroll across all inner loops. This yielded a runtime of 0.0195 ms, a ~7% improvement.

---

> > ### Author Rebuttal · Reviewer_XcbN · 2026-04-04
> >
> > Thank you for a detailed rebuttal.
> >
> > My main concerns were on (A) the generality (possible extension to Triton), (B) limited scope (isolated kernel development vs end-to-end optimization), (C) balancing between LLM generation and templated kernels, and (D) regarding better presentation of mutation hints.
> > My concerns on A and D are fully resolved. However, my concerns remain for B and C.
> >
> > Regarding B, depending on which kernel, input/output layout you select even if you get best performance on a kernel, it might not translate to end-to-end performance. Can you comment on how the work can be extended to address this issue?
> >
> > Regarding C, thank you for the experiments.
> > While I do agree with the overall message "We concluded that it is difficult to combine these two parts, and that it is better to do algorithmic optimization first, and to do parameter optimization as a second step, as done in the experiments reported in the paper."
> > It would be great to provide a little more detailed insights about the experiment. Hopefully a little in-depth analysis.
> > > With templated option: 48% correct, avg speedup of 2.4
> > > Without templated option: 52% correct, avg speedup of 2.9
> >
> > Also, it would be great to see how this might be for different kernels. Maybe some kernels benefit more from one direction over the other?

---

> > > ### Author Response · Authors · 2026-04-06
> > >
> > > Thanks for your response and interesting questions.
> > >
> > > **Regarding B** Our system optimizes kernels with respect to whatever benchmarking function the user provides. If the user supplies an end-to-end benchmark (e.g., a full model forward pass), the system optimizes within that full context, naturally capturing layout mismatches and reformatting overhead. This is what we already did in the Llama 3 example in the paper; we benchmarked a full forward pass. The scope of optimization is entirely user-driven.
> > >
> > > We acknowledge, however, that end-to-end benchmarking introduces practical challenges: model loading overhead and environment setup significantly increase test-time cost per iteration, which is expensive given the many evaluations our system performs. We are actively working to address this by supporting dockerized environments and datasets passed to the test worker, allowing the environment to be loaded once and reused across evaluations. We will add a discussion of this trade-off and our ongoing work to the paper.
> > >
> > > **Regarding C** Apologies for the brevity in the rebuttal. Detailed per-operation results are shown in the table below. The column "as second step" corresponds to results reported in the paper (Appendix Table 9, Table 2); the column "during training" shows the speedup for parameter optimization intermingled with algorithmic optimization.
> > >
> > > |parameter optimization                                                                  | as second step | during training
> > > | -----------------------------------------------| ----------------------| --------------------|
> > > | 16_ConvTranspose2d_Mish_Add_Hardtanh_Scaling.py   |                      1.785942       |           1.774603
> > > | 1_Conv2D_ReLU_BiasAdd.py                                            |    2.916996              |    2.764045
> > > | 21_Conv2d_Add_Scale_Sigmoid_GroupNorm.py               |                 3.045113        |          3.045113
> > > | 24_Conv3d_Min_Softmax.py                                       |         1.207178            |      1.211129
> > > | 32_Conv2d_Scaling_Min.py                                         |       7.870895           |       5.991379
> > > | 35_Conv2d_Subtract_HardSwish_MaxPool_Mish.py         |                   1.972043      |            1.879098
> > > | 37_Matmul_Swish_Sum_GroupNorm.py                             |           1.927711           |       0.774194
> > > | 46_Conv2d_Subtract_Tanh_Subtract_AvgPool.py                |             5.005025         |         3.730337
> > > | 5_ConvTranspose2d_Subtract_Tanh.py                            |          1.261731           |       1.229675
> > >
> > > Regarding whether specific kernels benefit more from one direction: the results are consistently inferior when intermingling – in many cases by a small margin or on par, but in some cases significantly. Parameter optimization during training thus never helps, but occasionally hurts.
> > >
> > > To understand why, we analyzed the generated kernels from both runs. In "during-training" mode, 52% of generated kernels are templated (used for parameter optimization), and of the remaining non-templated kernels, only 11% are correct. This suggests that when given the option, the LLM gravitates toward templated kernels early in the evolutionary process, focusing on parameter tuning at the expense of exploring algorithmic strategies, and subsequently struggling to produce correct non-templated kernels.

---

### Official Review · Reviewer_aQvs · 2026-03-13

**Soundness:** 3
**Presentation:** 3
**Significance:** 3
**Originality:** 3
**Overall Recommendation:** 5
**Confidence:** 4

**Summary:**

The authors present KernelFoundry, a hardware-aware evolutionary framework for optimizing GPU kernels using LLMs. There are 3 main points on which the paper is hinged. They implement a quality-diversity search using the MAP-Elites style with kernel-specific descriptors. They also use a meta-prompt evolution to co-evolve prompts with candidate kernels and mitigate context degradation. They also use templated kernel parameter optimization for hardware-dependent values such as block and tile sizes. They evaluate on KernelBench and robust-kbench and show on CUDA and SYCL kernels.

**Compliance With Llm Reviewing Policy:**

Affirmed.

**Final Justification:**

Most of my major concerns are addressed.

**Key Questions For Authors:**

Could you elaborate on the exact implementation of the static pattern matching used for the MAP-Elites classifier? Is this done via Clang AST traversal, or regex? How do you handle edge cases where the LLM implements a strategy (like shared memory tiling) using unconventional syntax that evades the pattern matcher?
What is the average token usage and corresponding API cost required to optimize a single kernel through the full 40 generations?
How sensitive is the meta-prompt evolution to the update frequency? You use a default of every 10 generations; did you experiment with continuous updates or single-shot reflections, and how did they impact diversity?
See Weakness.

**Limitations:**

Yes

**Strengths And Weaknesses:**

Strengths:
The MAP-Elites framing is a meaningful step beyond standard agent loops.
The inclusion of SYCL is a genuine plus. The paper’s claim that prior work has focused almost entirely on NVIDIA’s ecosystem is broadly fair, and expanding to an open, vendor-agnostic abstraction makes the work more interesting than a purely CUDA-only optimization paper.
The Llama 3 rotary embedding case study is a helpful attempt to show utility beyond benchmark-style toy tasks. Even if limited, it is more compelling than papers that stay entirely inside standardized synthetic evaluations.

Weakness:
The paper states that behavioral coordinates are computed deterministically from generated code via static pattern matching on SYCL and CUDA constructs. Static pattern matching can be brittle when applied to LLM-generated code, which may introduce unconventional syntax, custom wrapper functions, or aliases.
As demonstrated in Appendix G, utilizing a 20B open-source model yields multiple compilation failures and significantly lower performance. The framework is currently bottlenecked by the reasoning capabilities of very expensive, top-tier frontier models (GPT-4.1, Sonnet 3.5, o3-mini).
Running an evolutionary algorithm with LLMs in the loop is resource-intensive. With 40 generations and a population size of 8, plus the overhead of the meta-prompter LLM, the token usage could be immense. The paper would benefit greatly from reporting the average token consumption and monetary cost to optimize a single task.

---

> ### Author Rebuttal · Authors · 2026-03-30
>
> Thank you for the positive feedback! We answer to questions and concerns in the following.
>
> **[W1] Static pattern matching**
>
> We are using a regex-based heuristic for computing the grid coordinates. For each coordinate we try to match a set of weighted regular expressions and compute a confidence score. For example, static shared memory declarations in CUDA using the __shared__ keyword have a higher confidence than dynamic declarations for which we cannot check the allocation size with regular expressions and cannot be sure that shared memory is actually used. If the score exceeds a threshold, we assume that the respective optimization is present. We experimented with the libclang AST for SYCL code analysis but use regex for simplicity and faster parsing. However, analyzing the AST would not solve problems like the dynamic allocation of shared memory or matching dead code that cannot be removed by the compiler. To tackle this, we plan to use the profiler, which measures shared memory accesses and the type of executed instructions, in combination with static code analysis for more robust classification in future versions of our framework. For KernelBench problems -- where the generated code is a single translation unit -- we did not observe problems with our current implementation.
>
> **[W2] bottlenecked by large expensive models**
>
> We agree that the framework currently relies on top-tier models, which is a limitation shared across most approaches in the kernel generation field. The weaker results in Appendix G are partly due to limited familiarity with the latest SYCL API in open-source models, a pattern observed consistently across GPT-OSS, CodeLlama, and StarCoder. While CUDA and Triton may be easier to optimize with open-source models given their broader representation in training data, related work in this space also relies predominantly on large proprietary models due to their significantly stronger code generation capabilities.
> We see fine-tuned open-source models as a natural next step for KernelFoundry, and plan to explore this direction in future work to reduce dependency on closed models and improve SYCL-specific performance through targeted training.
>
> **[W3] Missing cost estimation**
>
> We agree that cost is an important factor. We measured the token usage for five KernelBench level 2 problems with the same setting as in the experiment in Table 1 (40 iterations, 4 branches, o3-mini). We find that, on average, one kernel requires 11102 input tokens and 5377 output tokens. Using the values from the GPT (https://developers.openai.com/api/docs/pricing), o3-mini has a cost of  \\$1.10 per 1M inputs and \\$4.40 per 1M output tokens. For 160 kernels (40 iterations x 4), this would result in a cost of 11102 * 160 * 1.1 / 1000000 + 5377 * 4.4 / 1000000 = 5.74. Thus, for o3-mini generating one kernel would cost \\$5.74. Using the latest models (gpt-5.4), the cost would be \\$17.34.
>
> The costs are comparable to related work, e.g. robust-kbench reports \\$5 API cost per kernel (and 6 H100 GPU hours) and METR shows convergence curves up to 20\\$ per kernel for o3-mini. (https://metr.org/blog/2025-02-14-measuring-automated-kernel-engineering/). Furthermore, much higher costs are justified if a tool can take substantial work from well-paid engineers who do kernel generation manually.
>
> The frequency of meta-prompting can be configured; for a frequency of 10 the cost would increase by 10\%. Regarding computational cost, note that our distributed pipeline minimizes GPU usage since only testing is executed on GPU.
> We will add this discussion to the paper.
>
> **[Q1] Implementation of pattern matching** - See W1
>
> **[Q2] Average token usage and API cost**- See W3
>
> **[Q3] How sensitive is the meta-prompt evolution to the update frequency?**
>
> Thank you for this interesting question. We have added additional results on meta-prompting in https://anonymous.4open.science/#!/r/kernelfoundry_rebuttal-79EE. In the limited time of the rebuttal, we could only run 20 iterations (population=4) with different configurations. The main results are provided in *table_prompt_evolution*, showing that higher update frequency indeed leads to additional performance gains. However, the impact of prompt evolution over 20 iterations is modest in general as prompt evolution should mainly reduce context degradation over many trials. A qualitative example is shown in Figure *prompt_evolution_100_trials*. See our response to reviewer TdVw Q2 for an analysis of how the prompts evolve. We will add these results to the paper.

---

> > ### Author Rebuttal · Reviewer_aQvs · 2026-04-05
> >
> > Thank you for the rebuttal. It has addressed most of my concerns.

---

### Official Review · Reviewer_TdVw · 2026-03-13

**Soundness:** 2
**Presentation:** 3
**Significance:** 3
**Originality:** 3
**Overall Recommendation:** 5
**Confidence:** 4

**Summary:**

The paper proposes KernelFoundry, an evolutionary framework for GPU kernel optimization that co-evolves prompts and kernels. It adapts MAP-Elites to kernel generation with a three-dimensional behavioral feature space capturing memory access patterns, algorithmic structure, and parallelism strategies. The system combines evolutionary search, meta-prompt evolution, gradient-guided mutation hints, and templated kernel parameter tuning. Experiments on filtered KernelBench and robust-kbench show consistent improvements over prior approaches.

**Compliance With Llm Reviewing Policy:**

Affirmed.

**Final Justification:**

The attached new results address my concerns.

**Key Questions For Authors:**

Q1. The experiments use only 40 evolutionary iterations, which seems relatively small for an evolutionary optimization process. How does the performance evolve beyond 40 iterations, and does the method eventually reach a clear saturation point?

Q2. Have the authors analyzed the trajectory of prompt evolution during the search process? For example, are there recurring patterns or phases in how prompts evolve? Such analysis could help extract useful strategies for designing a static multi-phase pipeline and may also shed light on the limitations of the self-evolving prompt approach.

**Limitations:**

Yes

**Strengths And Weaknesses:**

### strength

(1) Co-evolving prompts with kernels helps mitigate context pollution during long optimization runs and enables the system to adaptively switch optimization strategies as search progresses.

(2) Gradient-guided selection leverages historical parent–child transitions to estimate promising search directions, allowing the system to learn from prior exploration and guide kernel mutations more effectively.

### weakness

(1) Limited number of iterations. The experiments use only around 40 evolutionary iterations, which seems relatively small for an evolutionary search process. It is unclear when the performance would saturate or how the method behaves with longer runs.

(2) Limited analysis of key components. The paper introduces prompt co-evolution and gradient-informed selection as important components, but their individual contributions are unclear due to the lack of ablation studies. More detailed analysis would help demonstrate their effectiveness and clarify how much each component contributes to the overall improvement.

---

> ### Author Rebuttal · Authors · 2026-03-30
>
> Thank you for the positive evaluation! We address the remaining concerns in the following. Additional figures and tables are provided at https://anonymous.4open.science/#!/r/kernelfoundry_rebuttal-79EE.
>
> **[W1] Limited number of iterations**
>
> This is an important point. We did run KernelFoundry for 100 iterations on a set of 20 L2 KernelBench problems (SYCL) in an earlier experiment (repeating the experiments from the paper with 100 iterations was not feasible in the time of the rebuttal). As shown in Figure *normalized_100_trials* (see anonymous GitHub link above) some tasks converge early, while others still exhibit meaningful improvements even after 80 trials. We note that the min-max normalized view in Figure X can be misleading - some late jumps correspond to modest absolute gains; Figure *absolute_100_trials* (same link) shows the absolute speedups for reference.
> We believe this demonstrates that our method maintains sufficient diversity to avoid local minima and continue improving over many iterations. We acknowledge that the 40-iteration results reported in the paper are not the best achievable; however, we consider it important to report results under a controlled, fixed compute budget to ensure fair comparison with related work. In practice, users can straightforwardly increase the number of iterations to obtain further gains.
>
> **[W2] Missing ablation studies**
>
> Thank you for the feedback. We will add ablation studies to the paper. To get preliminary results within the short time of the rebuttal, we ran ablation studies on the L2 KernelBench subset (20 problems) for 20 iterations each. For results, see *table_ablation* in the shared folder linked above. Leaving out gradient-informed selection leads to a considerable performance drop, while changes to prompt evolution show moderate impact in this experiment. However, the advantages of prompt evolution are most prominent when running more iterations. We tested on just two tasks (the ones with largest speedup) how the speedup evolves over 100 iterations; Figure *prompt_evolution_100trials* shows the convergence with and without prompt evolution. With prompt evolution, the speedup increases faster and there are further jumps in performance in late iterations.
>
> **[Q1]**
>
> See W1
>
> **[Q2] Trajectory of prompt evolution**
>
> Thanks for this interesting question. We have done an additional analysis where we analyzed the changes of prompts generated during training. In Figure *prompt_similarity*, you can see their pairwise similarities, as measured with difflib’s SequenceMatcher, for a few randomly selected problems. While the prompt changes a lot for some problems, there are clusters in others where they remain similar. Also, there seem to be recurring patterns, as there are similarities between the first generated prompt and prompts generated much later in training. On manual inspection, there are more changes to “optimization strategies” and “common pitfalls” than for the sections “optimization philosophy” and “analysis guidance” (see section 3.5 in the paper). Observed additions to the prompt include 1) error correction guidance, e.g. “Reinterpreting host pointers to vector types inside kernels is unsafe unless the pointer is device-USM and properly aligned. Anti-pattern seen in generated code: reinterpret_cast<sycl::float4*>(conv_out_ptr)[i]. Fix: …” or 2) problem-specific optimization strategies such as “Use direct conv with shared memory when kernel sizes are small (3×3, 5×5) and memory reuse within a tile is high.”
>
> We believe that this design cannot be replaced easily by a static multi-phase pipeline, since a static pipeline could not insert problem-specific tips. Limitations are that the evolvable prompt parts are sometimes fully rewritten; future work should try to identify successful parts and retain those.
>
> We will add this additional analysis to the paper.

---

> > ### Author Rebuttal · Reviewer_TdVw · 2026-04-02
> >
> > The attached new results address my concerns.

---

### Decision · Program_Chairs · 2026-04-30

**Decision:**

Accept (regular)

**Comment:**

The paper presents KernelFoundry, a hardware-aware evolutionary framework for GPU kernel optimization that combines quality-diversity search, prompt co-evolution, gradient-informed mutation guidance, and template-based parameter tuning. The reviewers were uniformly positive on the overall contribution, highlighting the novelty of the framework, the relevance of the problem, and the value of supporting SYCL in addition to CUDA. The work introduces a compelling framework that others are likely to build on, especially at the intersection of LLM-based code generation, evolutionary search, and hardware-aware optimization. The authors are encouraged to incorporate their responses in the rebuttal into the final version of the paper.